# SimpleMem: Efficient Lifelong Memory for LLM Agents

Jiaqi Liu [1] [*]  Yaofeng Su [1] [*]  Peng Xia [1]  Siwei Han [1]
Zeyu Zheng [2]  Cihang Xie [3]  Mingyu Ding [1]  Huaxiu Yao [1]

## Abstract

To support long-term interaction in complex environments, LLM agents require memory systems that manage historical experiences. Existing approaches either retain full interaction histories via passive context extension, leading to substantial redundancy, or rely on iterative reasoning to filter noise, incurring high token costs. To address this challenge, we introduce SimpleMem, an efficient memory framework based on semantic lossless compression. We propose a three-stage pipeline designed to maximize information density and token utilization: (1) Semantic Structured Compression, which distills unstructured interactions into compact, multi-view indexed memory units; (2) Online Semantic Synthesis, an intra-session process that instantly integrates related context into unified abstract representations to eliminate redundancy; and (3) Intent-Aware Retrieval Planning, which infers search intent to dynamically determine retrieval scope and construct precise context efficiently. Experiments on benchmark datasets show that our method consistently outperforms baseline approaches in accuracy, retrieval efficiency, and inference cost, achieving an average F1 improvement of 26.4% in LoCoMo while reducing inference-time token consumption by up to 30×, demonstrating a superior balance between performance and efficiency. Code is available at https://github.com/aiming-lab/SimpleMem.

## 1. Introduction

Large Language Model (LLM) agents have recently demonstrated remarkable capabilities across a wide range of tasks (Xia et al., 2025; Team et al., 2025; Qiu et al., 2025).

However, constrained by fixed context windows, existing agents exhibit significant limitations when engaging in long-context and multi-turn interaction scenarios (Liu et al., 2023; Wang et al., 2024a; Liu et al., 2025; Hu et al., 2025; Tu et al., 2025). To facilitate reliable long-term interaction, LLM agents require robust memory systems to efficiently manage and utilize historical experience (Dev & Taranjeet, 2024; Fang et al., 2025; Wang & Chen, 2025; Tang et al., 2025; Yang et al., 2025; Ouyang et al., 2025).

While recent research has extensively explored the design of memory modules for LLM agents, current systems still suffer from suboptimal retrieval efficiency and low token utilization (Fang et al., 2025; Hu et al., 2025). On one hand, many existing systems maintain complete interaction histories through full-context extension (Li et al., 2025; Zhong et al., 2024). However, this approach introduce substantial redundant information (Hu et al., 2025). Specifically, during long-horizon interactions, user inputs and model responses accumulate substantial low-entropy noise (e.g., repetitive logs, non-task-oriented dialogue), which degrades the effective information density of the memory buffer. This redundancy adversely affects memory retrieval and downstream reasoning, often leading to middle-context degradation phenomena (Liu et al., 2023), while also incurring significant computational overhead during retrieval and secondary inference. On the other hand, some agentic frameworks mitigate noise through online filtering based on iterative reasoning procedures (Yan et al., 2025; Packer et al., 2023). Although such approaches improve retrieval relevance, they rely on repeated inference cycles, resulting in substantial computational cost, including increased latency and token usage. As a result, neither paradigm achieves efficient allocation of memory and computation resources.

To address these limitations, we introduce **SimpleMem**, an efficient memory framework inspired by the Complementary Learning Systems (CLS) theory (Kumaran et al., 2016) and built around structured semantic compression. The objective of SimpleMem is to improve information efficiency under fixed context and token budgets. We develop a three-stage pipeline that supports dynamic memory compression, organization, and adaptive retrieval: (1) **Semantic Structured Compression**: we apply a semantic density gating mechanism via LLM-based qualitative assessment. The

[*]Equal contribution  [1]UNC-Chapel Hill [2]University of California, Berkeley [3]University of California, Santa Cruz. Correspondence to: Jiaqi Liu <jqliu@cs.unc.edu>, Mingyu Ding <md@cs.unc.edu>, Huaxiu Yao <huaxiu@cs.unc.edu>.

*Proceedings of the 43$^{rd}$ International Conference on Machine Learning*, Seoul, South Korea. PMLR 306, 2026. Copyright 2026 by the author(s).

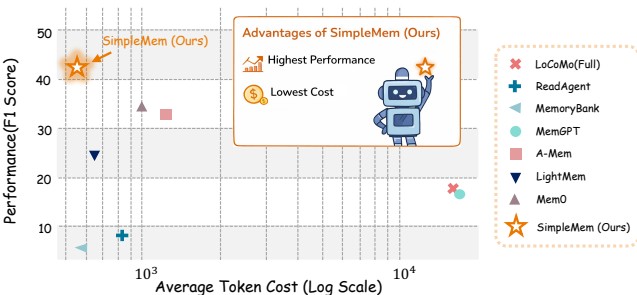

*Figure 1.* Performance vs. Efficiency Trade-off. Comparison of F1 against Token Cost on the LoCoMo benchmark. SimpleMem achieves high accuracy with minimal token consumption.

system uses the foundation model as a semantic judge to estimate information gain relative to history, preserving only content with high downstream utility. Retained information is reformulated into compact memory units and indexed jointly using dense semantic embeddings, sparse lexical features, and symbolic metadata. (2) **Online Semantic Synthesis**: inspired by biological consolidation and optimized for real-time interaction, we introduce an intra-session process that reorganizes memory on-the-fly. Related memory units are synthesized into higher-level abstract representations during the write phase, allowing repetitive or structurally similar experiences to be denoised and compressed immediately. (3) **Intent-Aware Retrieval Planning**: we employ a planning-based retrieval strategy that infers latent search intent to determine retrieval scope dynamically. The system constructs a precise context by querying multiple indexes (symbolic, semantic, lexical) and unifying results through ID-based deduplication, balancing structural constraints and semantic relevance without complex linear weighting.

Our primary contribution is SimpleMem, an efficient memory framework grounded in structured semantic compression, which improves information efficiency through principled memory organization, online synthesis, and intent-aware planning. As shown in Figure 1, our empirical experiments demonstrate that SimpleMem establishes a new state-of-the-art with an F1 score, outperforming strong baselines like Mem0 by 26.4%, while reducing inference token consumption by 30× compared to full-context models.

## 2. The SimpleMem Architecture

In this section, we present **SimpleMem**, which operates through a three-stage pipeline (see Figure 2 for the detailed architecture). Specifically, we first describe the *Semantic Structured Compression*, which utilizes implicit semantic gating to filter redundant interaction content and reformulate raw dialogue streams into compact memory units. Next, we describe *Online Semantic Synthesis*, an on-the-fly mechanism that instantly synthesizes related memory units into higher-level abstract representations, ensuring a compact and noise-free memory topology. Finally, we present *Intent-*

*Aware Retrieval Planning*, which infers latent search intent to dynamically adjust retrieval scope, constructing precise and token-efficient contexts for downstream reasoning.

### 2.1. Semantic Structured Compression

A primary bottleneck in long-term interaction is *context inflation*, the accumulation of raw, low-entropy dialogue. For example, a large portion of interaction segments in the real-world consists of phatic chit-chat or redundant confirmations, which contribute little to downstream reasoning but consume substantial context capacity. To address this, we introduce a mechanism to actively filter and restructure information at the source.

Specifically, first, incoming dialogue is segmented into overlapping sliding windows $W$ of fixed length, where each window represents a short contiguous span of recent interaction. These windows serve as the basic units for processing.

Unlike traditional approaches that rely on rigid heuristic filters or separate classification models, we employ an implicit semantic density gating mechanism integrated directly into the generation process. We model the information assessment as an instruction-following task performed by the foundation model itself. The system leverages the attention mechanism of the LLM $f$ to identify high-entropy spans within the window $W$ relative to the immediate history $H$.

Formally, we define the gating function $\Phi_{\text{gate}}$ not as a binary classifier, but as a generative filter resulting from the model's extraction capability:

$$\Phi_{\text{gate}}(W) \to \{m_k\} \quad \text{s.t.} \quad |\{m_k\}| \geq 0 \quad (1)$$

Here, the generation of an empty set ($\emptyset$) inherently signifies a low-density window (e.g., pure phatic chitchat), effectively discarding it without explicit threshold tuning. This instruction-driven gating allows the system to capture subtle semantic nuances while naturally filtering redundancy through the model's semantic compression objectives.

For windows containing valid semantic content, the system performs a unified *De-linearization Transformation* $\mathcal{F}_\theta$. Instead of sequential independent modules, we optimize the extraction, coreference resolution, and temporal anchoring as a joint generation task. The transformation projects the raw dialogue window $W$ directly into a set of context-independent memory units $\{m_k\}$:

$$\{m_k\} = \mathcal{F}_\theta(W; H) \approx (g_{\text{time}} \circ g_{\text{coref}} \circ g_{\text{ext}})(W). \quad (2)$$

In this unified pass, the model follows strict instructional constraints to: (1) resolve ambiguous pronouns to specific entity names ($g_{\text{coref}}$), (2) convert relative temporal expressions into absolute ISO-8601 timestamps ($g_{\text{time}}$), and (3) atomize complex dialogue flows into self-contained factual

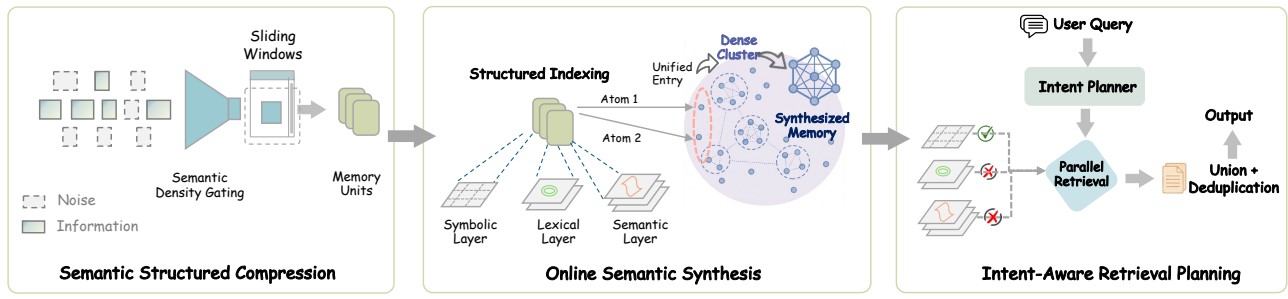

*Figure 2.* The SimpleMem Architecture. SimpleMem follows a three-stage pipeline: (1) Semantic Structured Compression filters low-utility dialogue and converts informative windows into compact, context-independent memory units. (2) Online Semantic Synthesis consolidates related fragments during writing, maintaining a compact and coherent memory topology. (3) Intent-Aware Retrieval Planning infers search intent to adapt retrieval scope and query forms, enabling parallel multi-view retrieval and token-efficient context construction.

statements. By aggregating all resulting units $m_k$ across sliding windows, we obtain the complete memory set $\mathcal{M}$.

Following compression, the system organizes the memory units to support storage and retrieval. This stage consists of two synergistic processes: (i) structured multi-view indexing for precise access, and (ii) online semantic synthesis for minimizing redundancy at the point of creation.

To support flexible and high-fidelity retrieval, each memory unit is indexed through three complementary representations. First, at the *Semantic Layer*, we map the entry to a dense vector space $s_k$ using embedding models, capturing abstract meaning to enable fuzzy matching (e.g., retrieving "latte" when querying "hot drink"). Second, the *Lexical Layer* utilizes an inverted index-based sparse representation. This acts as a high-dimensional sparse vector $l_k$ focusing on exact keyword matches and rare proper nouns, ensuring that specific entities are not diluted in dense vector space. Third, the *Symbolic Layer* extracts structured metadata, such as timestamps and entity types, to enable deterministic filtering logic. Formally, for a given memory unit $m_k$, these projections form the comprehensive indexing $\mathcal{I}$:

$$\mathcal{I}(m_{t,k}) = \begin{cases} s_k = E_{\text{dense}}(m_k) & \text{(Semantic Layer)} \\ l_k = E_{\text{sparse}}(m_k) & \text{(Lexical Layer)} \\ r_k = E_{\text{sym}}(m_k) & \text{(Symbolic Layer)} \end{cases} \quad (3)$$

This architecture allows the system to flexibly query information based on conceptual similarity, exact keyword matches, or structured metadata constraints.

### 2.2. Online Semantic Synthesis

While this multi-view indexing strategy facilitates access, naively accumulating raw extractions leads to fragmentation, causing the memory structure to grow in a purely additive and unregulated manner that fails to adapt in real time to the evolving semantic context of an ongoing interaction. To address this, we introduce *Online Semantic Synthesis*, an intra-session consolidation mechanism. Unlike traditional systems that rely on asynchronous background maintenance,

SimpleMem performs synthesis on-the-fly during the write phase. The model analyzes the stream of extracted facts within the current session scope and synthesizes related fragments into unified, high-density entries before they are committed to the database.

Formally, we define this synthesis as a transformation function $\mathcal{F}_{\text{syn}}$ that maps a set of new observations $O_{\text{session}}$ to a consolidated memory entry $\mathcal{F}_{\text{syn}}(O_{\text{session}}, \mathcal{C}_{\text{context}}; f)$, where $\mathcal{C}_{\text{context}}$ represents the current conversational context. This operation denoises the input by merging scattered details into a coherent whole. For instance, rather than storing three separate fragments like `"User wants coffee"`, `"User prefers oat milk"`, and `"User likes it hot"`, the synthesis layer consolidates them into a single, comprehensive entry: `"User prefers hot coffee with oat milk"`. This proactive synthesis ensures that the memory topology remains compact and free of redundant fragmentation, significantly reducing the burden on the retrieval system during future interactions.

### 2.3. Intent-Aware Retrieval Planning

After memory entries are organized, the final challenge is to retrieve relevant information under constrained context budgets. Standard retrieval approaches typically fetch a fixed number of entries, which often results in recall failure for complex queries or token wastage for simple ones. To address this, we introduce *Intent-Aware Retrieval Planning*, a mechanism that dynamically determines the retrieval scope and depth by inferring the user's latent search intent.

Unlike systems that rely on scalar complexity classifiers, SimpleMem leverages the reasoning capabilities of the LLM to generate a comprehensive retrieval plan. Given a query $q$ and history $H$, the planning module $\mathcal{P}$ acts as a reasoner to decompose the information needs and estimate the necessary search depth $d$:

$$\{q_{\text{sem}}, q_{\text{lex}}, q_{\text{sym}}, d\} \sim \mathcal{P}(q, H) \quad (4)$$

where $q_{\text{sem}}, q_{\text{lex}}$, and $q_{\text{sym}}$ are optimized queries for semantic,

lexical, and symbolic retrieval respectively. The parameter $d$ represents the *adaptive retrieval depth*, which reflects the estimated complexity of the query. Based on $d$, the system utilizes a candidate limit $n$ (where $n \propto d$) to balance recall coverage against context window constraints.

Guided by this plan, the system executes a parallel multi-view retrieval. We simultaneously query all three index layers defined in Section 2.1, imposing the quantity limit $n$ on each path:

$$\mathcal{R}_{\text{sem}} = \text{Top-}n(\cos(E(q_{\text{sem}}), E(m_i)) \mid m_i \in \mathcal{M})$$
$$\mathcal{R}_{\text{lex}} = \text{Top-}n(\text{BM25}(q_{\text{lex}}, m_i) \mid m_i \in \mathcal{M}) \quad (5)$$
$$\mathcal{R}_{\text{sym}} = \text{Top-}n(\{m_i \in \mathcal{M} \mid \text{Meta}(m_i) \models q_{\text{sym}}\})$$

Here, each view captures distinct relevance signals: $\mathcal{R}_{\text{sem}}$ retrieves based on dense embedding similarity; $\mathcal{R}_{\text{lex}}$ matches exact keywords or proper nouns; and $\mathcal{R}_{\text{sym}}$ filters entries based on structured metadata constraints.

Finally, we construct the context $\mathcal{C}_q$ by merging the results from these three views using a set union operation. This step naturally deduplicates overlapping entries, ensuring a comprehensive yet compact context:

$$\mathcal{C}_q = \mathcal{R}_{\text{sem}} \cup \mathcal{R}_{\text{lex}} \cup \mathcal{R}_{\text{sym}} \quad (6)$$

This hybrid approach ensures that strong signals from any view are preserved, allowing the system to adaptively scale its retrieval volume $n$ based on the inferred depth $d$.

## 3. Experiments

In this section, we evaluate SimpleMem on the benchmark to answer the following research questions: (1) Does SimpleMem outperform other memory systems in complex long-term reasoning understanding tasks? (2) Can SimpleMem achieve a superior trade-off between retrieval accuracy and token consumption? (3) How effective are the proposed components? (4) What factors account for the observed performance and efficiency gains?

### 3.1. Experimental Setup

**Benchmark Dataset.** We evaluate performance on the LoCoMo (Maharana et al., 2024) and LongMemEval-S (Wu et al., 2024) benchmarks. Brief descriptions are provided below, with additional details in Appendix B.

*LoCoMo* (Maharana et al., 2024) is specifically designed to test the limits of LLMs in processing long-term conversational dependencies. The dataset comprises conversation samples ranging from 200 to 400 turns, containing complex temporal shifts and interleaved topics. The evaluation set consists of 1,986 questions categorized into four distinct reasoning types: (1) multi-hop reasoning; (2) temporal reasoning; (3) open Domain; (4) single hop.

*LongMemEval-S* (Wu et al., 2024) features extreme context lengths that pose severe challenges for memory systems. Unlike standard benchmarks, it requires precise answer localization across multiple sub-categories (e.g., temporal events, user preferences) within exceptionally long interaction histories. We use gpt-4.1-mini to evaluate answer correctness against ground-truth references, labeling responses as CORRECT or WRONG based on semantic and temporal alignment. The full evaluation prompt is provided in Appendix A.4.

**Baselines.** We compare SimpleMem with representative memory-augmented systems: LOCOMO (Maharana et al., 2024), READAGENT (Lee et al., 2024), MEMORY-BANK (Zhong et al., 2024), MEMGPT (Packer et al., 2023), A-MEM (Xu et al., 2025), LIGHTMEM (Fang et al., 2025), and Mem0 (Dev & Taranjeet, 2024).

**Backbone Models.** To test robustness across capability scales, we instantiate each baseline and SimpleMem on multiple LLM backends: GPT-4o, GPT-4.1-mini, Qwen-Plus, Qwen2.5 (1.5B/3B), and Qwen3 (1.7B/8B).

**Implementation Details.** For semantic structured compression, we use a sliding window of size $W = 20$. Memory indexing is implemented using LanceDB with a multi-view design: Qwen3-embedding-0.6b (1024 dimensions) for dense semantic embeddings, BM25 for sparse lexical indexing, and SQL-based metadata storage for symbolic attributes. During retrieval, we employ adaptive query-aware retrieval, where the retrieval depth is dynamically adjusted based on estimated query complexity, ranging from $k_{\min} = 3$ for simple lookups to $k_{\max} = 20$ for complex reasoning queries.

**Evaluation Metrics.** For LoCoMo, we report F1 and BLEU-1 (accuracy), Adversarial Success Rate (robustness to distractors), and Token Cost (retrieval efficiency). For LongMemEval-S, we use its standard accuracy-style metric.

### 3.2. Results and Analysis

Tables 1 and 3 present detailed performance comparisons on the LoCoMo benchmark across different model scales, while Table 2 reports results on LongMemEval-S.

**Performance on High-Capability Models.** Across LoCoMo and LongMemEval-S, SimpleMem consistently outperforms existing memory systems across model scales, achieving strong and robust gains in accuracy.

On LoCoMo (Table 1), SimpleMem leads all baselines. Using GPT-4.1-mini, it achieves an Average F1 of 43.24, substantially exceeding Mem0 (34.20) and the full-context baseline (18.70). The largest gains are observed in Temporal Reasoning, where SimpleMem reaches 58.62 F1 compared to 48.91 for Mem0, underscoring the effectiveness of seman-

*Table 1.* Performance on the LoCoMo benchmark with High-Capability Models (GPT-4.1 series and Qwen3-Plus). SimpleMem achieves superior efficiency-performance balance.

| Model | Method | MultiHop F1 | MultiHop BLEU | Temporal F1 | Temporal BLEU | OpenDomain F1 | OpenDomain BLEU | SingleHop F1 | SingleHop BLEU | Average F1 | Average BLEU | Token Cost |
|-------|--------|-----|------|-----|------|-----|------|-----|------|-----|------|------|
| | LoCoMo | 25.02 | 21.62 | 12.04 | 10.63 | 19.05 | 17.07 | 18.68 | 15.87 | 18.70 | 16.30 | 16,910 |
| | ReadAgent | 6.48 | 5.6 | 5.31 | 4.23 | 7.66 | 6.62 | 9.18 | 7.91 | 7.16 | 6.09 | 643 |
| | MemoryBank | 5.00 | 4.68 | 5.94 | 4.78 | 5.16 | 4.52 | 5.72 | 4.86 | 5.46 | 4.71 | 432 |
| **GPT-4.1-mini** | MemGPT | 17.72 | 16.02 | 19.44 | 16.54 | 11.29 | 10.18 | 25.59 | 24.25 | 18.51 | 16.75 | 16,977 |
| | A-Mem | 25.06 | 17.32 | 51.01 | 44.75 | 13.22 | 14.75 | 41.02 | 36.99 | 32.58 | 28.45 | 2,520 |
| | LightMem | 24.96 | 21.66 | 20.55 | 18.39 | 19.21 | 17.68 | 33.79 | 29.66 | 24.63 | 21.85 | 612 |
| | Mem0 | 30.14 | 27.62 | 48.91 | 44.82 | 16.43 | 14.94 | 41.3 | 36.17 | 34.20 | 30.89 | 973 |
| | **SimpleMem** | **43.46** | **38.82** | **58.62** | **50.10** | **19.76** | **18.04** | **51.12** | **43.53** | **43.24** | **37.62** | 531 |
| | LoCoMo | 28.00 | 18.47 | 9.09 | 5.78 | 16.47 | 14.80 | 61.56 | 54.19 | 28.78 | 23.31 | 16,910 |
| | ReadAgent | 14.61 | 9.95 | 4.16 | 3.19 | 8.84 | 8.37 | 12.46 | 10.29 | 10.02 | 7.95 | 805 |
| | MemoryBank | 6.49 | 4.69 | 2.47 | 2.43 | 6.43 | 5.30 | 8.28 | 7.10 | 5.92 | 4.88 | 569 |
| **GPT-4o** | MemGPT | 30.36 | 22.83 | 17.29 | 13.18 | 12.24 | 11.87 | 40.16 | 36.35 | 25.01 | 21.06 | 16,987 |
| | A-Mem | 32.86 | 23.76 | 39.41 | 31.23 | 17.10 | 15.84 | 44.43 | 38.97 | 33.45 | 27.45 | 1,216 |
| | LightMem | 28.15 | 21.83 | 36.53 | 29.12 | 13.38 | 11.54 | 33.76 | 28.02 | 27.96 | 22.63 | 645 |
| | Mem0 | 35.13 | 27.56 | 52.38 | 44.15 | 17.73 | 15.92 | 39.12 | 35.43 | 36.09 | 30.77 | 985 |
| | **SimpleMem** | **35.89** | **32.83** | **56.71** | **20.57** | **18.23** | **16.34** | **45.41** | **39.25** | **39.06** | **27.25** | 550 |
| | LoCoMo | 24.15 | 18.94 | 16.57 | 13.28 | 11.81 | 10.58 | 38.58 | 28.16 | 22.78 | 17.74 | 16,910 |
| | ReadAgent | 9.52 | 6.83 | 11.22 | 8.15 | 5.41 | 5.23 | 9.85 | 7.96 | 9.00 | 7.04 | 742 |
| | MemoryBank | 5.25 | 4.94 | 1.77 | 6.26 | 5.88 | 6.00 | 6.90 | 5.57 | 4.95 | 5.69 | 302 |
| **Qwen3-Plus** | MemGPT | 25.80 | 17.50 | 24.10 | 18.50 | 9.50 | 7.80 | 40.20 | 42.10 | 24.90 | 21.48 | 16,958 |
| | A-Mem | 26.50 | 19.80 | 46.10 | 35.10 | 11.90 | 11.50 | 43.80 | 36.50 | 32.08 | 25.73 | 1,427 |
| | LightMem | 28.95 | 24.13 | 42.58 | 38.52 | 16.54 | 13.23 | 40.78 | 36.52 | 32.21 | 28.10 | 606 |
| | Mem0 | 32.42 | 21.24 | 47.53 | 39.82 | 17.18 | 14.53 | 46.25 | 37.52 | 35.85 | 28.28 | 1,020 |
| | **SimpleMem** | **33.74** | **29.04** | **50.87** | **43.31** | **18.41** | **16.24** | **46.94** | **38.16** | **37.49** | **31.69** | 583 |

*Table 2.* Performance comparison on the LongMemEval benchmark. The evaluation uses `gpt-4.1-mini` as the judge. SimpleMem achieves the best overall performance while maintaining balanced capabilities across different sub-tasks.

| Model | Method | Temporal | Multi-Session | Knowledge-Update | Single-Session-User | Single-Session-Assistant | Single-Session-Preference | Average |
|-------|--------|----------|---------------|------------------|---------------------|--------------------------|---------------------------|---------|
| | Full-context | 27.06% | 30.08% | 41.03% | 47.14% | 32.14% | 60.00% | 39.57% |
| **GPT-4.1-mini** | Mem0 | 40.60% | 50.37% | 69.23% | 87.14% | 48.21% | 63.33% | 59.81% |
| | LightMem | **85.71%** | 47.37% | **92.30%** | **88.57%** | 21.43% | 76.67% | 68.67% |
| | **SimpleMem** | 83.46% | **60.92%** | 79.48% | 85.71% | **75.00%** | **76.67%** | **76.87%** |
| | Full-context | 51.88% | 39.10% | 70.51% | 65.71% | 96.43% | 16.67% | 56.72% |
| **GPT-4.1** | Mem0 | 43.61% | 54.89% | 75.64% | 54.29% | 39.29% | **83.33%** | 58.51% |
| | LightMem | 84.96% | 57.89% | **89.74%** | 87.14% | 71.43% | 70.00% | 76.86% |
| | **SimpleMem** | **86.47%** | **81.20%** | 80.76% | **98.57%** | 76.79% | 80.00% | **83.97%** |

tic structured compression in resolving complex temporal dependencies. These improvements persist at larger scales: on GPT-4o, SimpleMem attains the highest Average F1 (39.06), outperforming Mem0 (36.09) and A-Mem (33.45).

A task-level breakdown on LoCoMo further highlights the balanced capabilities of SimpleMem. In SingleHop QA, SimpleMem consistently achieves the best performance (e.g., 51.12 F1 on GPT-4.1-mini), demonstrating precise factual retrieval. In more challenging MultiHop settings, SimpleMem significantly outperforms Mem0 and Light-Mem on GPT-4.1-mini, indicating its ability to bridge disconnected facts and support deep reasoning without relying on expensive iterative retrieval loops.

Results on LongMemEval-S (Table 2) further demonstrate the robustness of SimpleMem under extreme context lengths. Using the `gpt-4.1-mini` backbone, Simple-Mem achieves the highest average accuracy of 76.87%, outperforming LightMem (68.67%) and substantially exceed-

ing Mem0 (59.81%) and the full-context baseline (39.57%). Gains are most pronounced in the challenging *Multi-Session* category, where SimpleMem attains 60.92% accuracy, compared to 47.37% for LightMem and 30.08% for full-context, highlighting its effectiveness in cross-session information integration under severe context constraints.

When scaled to the more capable `gpt-4.1` backbone, SimpleMem maintains its state-of-the-art performance with an average accuracy of 83.97%. It is particularly strong in the *Single-Session User* task (98.57%), demonstrating near-perfect recall of immediate user-provided details. While LightMem exhibits strong performance on temporally specific queries, SimpleMem offers a more balanced profile: it avoids catastrophic failures in assistant-focused recall (where LightMem drop significantly on the mini model) while maintaining high accuracy in complex multi-session retrieval. This balance suggests that SimpleMem's structured indexing strategy effectively disentangles episodic

*Table 3.* Performance on the LoCoMo benchmark with Efficient Models (Small parameters). SimpleMem demonstrates robust performance even on 1.5B/3B models, often surpassing larger models using baseline memory systems.

| Model | Method | MultiHop | | Temporal | | OpenDomain | | SingleHop | | Average | | Token |
|---|---|---|---|---|---|---|---|---|---|---|---|---|
| | | F1 | BLEU | F1 | BLEU | F1 | BLEU | F1 | BLEU | F1 | BLEU | Cost |
| Qwen2.5-1.5b | LoCoMo | 9.05 | 6.55 | 4.25 | 4.04 | 9.91 | 8.50 | 11.15 | 8.67 | 8.59 | 6.94 | 16,910 |
| | ReadAgent | 6.61 | 4.93 | 2.55 | 2.51 | 5.31 | 12.24 | 10.13 | 7.54 | 6.15 | 6.81 | 752 |
| | MemoryBank | 11.14 | 8.25 | 4.46 | 2.87 | 8.05 | 6.21 | 13.42 | 11.01 | 9.27 | 7.09 | 284 |
| | MemGPT | 10.44 | 7.61 | 4.21 | 3.89 | 13.42 | 11.64 | 9.56 | 7.34 | 9.41 | 7.62 | 16,953 |
| | A-Mem | 18.23 | 11.94 | 24.32 | 19.74 | 16.48 | 14.31 | 23.63 | 19.23 | 20.67 | 16.31 | 1,300 |
| | LightMem | 16.43 | 11.39 | 22.92 | 18.56 | 15.06 | 11.23 | 23.28 | 19.24 | 19.42 | 15.11 | 605 |
| | Mem0 | 20.18 | 14.53 | 27.42 | 22.14 | 19.83 | 15.68 | 27.63 | 23.42 | 23.77 | 18.94 | 942 |
| | **SimpleMem** | **21.85** | **16.10** | **29.12** | **23.50** | **21.05** | **16.80** | **28.90** | **24.50** | **25.23** | **20.23** | 678 |
| Qwen2.5-3b | LoCoMo | 4.61 | 4.29 | 3.11 | 2.71 | 4.55 | 5.97 | 7.03 | 5.69 | 4.83 | 4.67 | 16,910 |
| | ReadAgent | 2.47 | 1.78 | 3.01 | 3.01 | 5.57 | 5.22 | 3.25 | 2.51 | 3.58 | 3.13 | 776 |
| | MemoryBank | 3.60 | 3.39 | 1.72 | 1.97 | 6.63 | 6.58 | 4.11 | 3.32 | 4.02 | 3.82 | 298 |
| | MemGPT | 5.07 | 4.31 | 2.94 | 2.95 | 7.04 | 7.10 | 7.26 | 5.52 | 5.58 | 4.97 | 16,961 |
| | A-Mem | 12.57 | 9.01 | **27.59** | **25.07** | 7.12 | 7.28 | 17.23 | 13.12 | 16.13 | 13.62 | 1,137 |
| | LightMem | 16.43 | 11.39 | 6.92 | 4.56 | 8.06 | 7.23 | 18.28 | 15.24 | 12.42 | 9.61 | 605 |
| | Mem0 | 16.89 | 11.54 | 8.52 | 6.23 | 10.24 | 8.82 | 16.47 | 12.43 | 13.03 | 9.76 | 965 |
| | **SimpleMem** | **17.03** | **11.87** | 21.47 | 19.50 | **12.52** | **10.19** | **20.90** | **18.01** | **17.98** | **14.89** | 572 |
| Qwen3-1.7b | LoCoMo | 10.28 | 8.82 | 6.45 | 5.78 | 10.42 | 9.02 | 11.16 | 10.35 | 9.58 | 8.49 | 16,910 |
| | ReadAgent | 7.50 | 5.60 | 3.15 | 2.95 | 6.10 | 12.45 | 10.80 | 8.15 | 6.89 | 7.29 | 784 |
| | MemoryBank | 11.50 | 8.65 | 4.95 | 3.20 | 8.55 | 6.80 | 13.90 | 11.50 | 9.73 | 7.54 | 290 |
| | MemGPT | 11.50 | 8.20 | 4.65 | 4.10 | 13.85 | 11.90 | 10.25 | 7.85 | 10.06 | 8.01 | 16,954 |
| | A-Mem | 18.45 | 11.80 | 25.82 | 18.45 | 10.90 | 9.95 | 21.58 | 16.72 | 19.19 | 14.23 | 1,258 |
| | LightMem | 14.84 | 11.56 | 9.35 | 7.85 | 13.76 | 10.59 | 28.14 | 22.89 | 16.52 | 13.22 | 679 |
| | Mem0 | 18.23 | 13.44 | 18.54 | 14.22 | 16.82 | 13.54 | 31.15 | 26.42 | 21.19 | 16.91 | 988 |
| | **SimpleMem** | **20.85** | **15.42** | **26.75** | **18.63** | **17.92** | **14.15** | **32.85** | **26.46** | **24.59** | **18.67** | 730 |
| Qwen3-8b | LoCoMo | 13.50 | 9.20 | 6.80 | 5.50 | 10.10 | 8.80 | 14.50 | 11.20 | 11.23 | 8.68 | 16,910 |
| | ReadAgent | 7.20 | 5.10 | 3.50 | 3.10 | 5.50 | 5.40 | 8.10 | 6.20 | 6.08 | 4.95 | 721 |
| | MemoryBank | 9.50 | 7.10 | 3.80 | 2.50 | 7.50 | 6.50 | 9.20 | 7.50 | 7.50 | 5.90 | 287 |
| | MemGPT | 14.20 | 9.80 | 5.50 | 4.20 | 12.50 | 10.80 | 11.50 | 9.10 | 10.93 | 8.48 | 16,943 |
| | A-Mem | 20.50 | 13.80 | 22.50 | 18.20 | 13.20 | 10.50 | 26.80 | 21.50 | 20.75 | 16.00 | 1,087 |
| | LightMem | 18.53 | 14.23 | 26.78 | 21.52 | 14.12 | 11.24 | 29.48 | 23.83 | 22.23 | 17.71 | 744 |
| | Mem0 | 22.42 | 16.83 | 32.48 | 26.13 | 15.23 | 12.54 | 33.05 | 27.24 | 25.80 | 20.69 | 1,015 |
| | **SimpleMem** | **28.97** | **24.93** | **42.85** | **36.49** | 15.35 | 13.90 | **46.62** | **40.69** | **33.45** | **29.00** | 621 |

noise from salient facts, providing a reliable memory substrate for long-term interaction.

**Token Efficiency.** A key strength of SimpleMem lies in its inference-time efficiency. As reported in the rightmost columns of Tables 1 and 3, full-context approaches such as LOCOMO and MEMGPT consume approximately 16,900 tokens per query. In contrast, SimpleMem reduces token usage by roughly 30×, averaging 530-580 tokens per query. Furthermore, compared to optimized retrieval baselines like Mem0 (∼980 tokens) and A-Mem (∼1,200+ tokens), SimpleMem reduces token usage by 40-50% while delivering superior accuracy. For instance, on GPT-4.1-mini, SimpleMem uses only 531 tokens to achieve state-of-the-art performance, whereas ReadAgent consumes more (643 tokens) but achieves far lower accuracy (7.16 F1). This validates the effectiveness of SimpleMem in strictly controlling context bandwidth without sacrificing information density.

**Performance on Smaller Models.** Table 3 highlights the ability of SimpleMem to empower smaller parameter models. On Qwen3-8b, SimpleMem achieves an impressive

Average F1 of 33.45, significantly surpassing Mem0 (25.80) and LightMem (22.23). Crucially, a 3B-parameter model (Qwen2.5-3b) paired with SimpleMem achieves 17.98 F1, outperforming the same model with Mem0 (13.03) by nearly 5 points. Even on the extremely lightweight Qwen2.5-1.5b, SimpleMem maintains robust performance (25.23 F1), beating larger models using inferior memory strategies (e.g., Qwen3-1.7b with Mem0 scores 21.19).

### 3.3. Efficiency Analysis

We conduct a comprehensive evaluation of computational efficiency, examining both end-to-end system latency and the scalability of memory indexing and retrieval. To assess practical deployment viability, we measured the full lifecycle costs on the LoCoMo-10 dataset using GPT-4.1-mini.

As illustrated in Table 4, SimpleMem exhibits superior efficiency across all operational phases. In terms of memory construction, our system achieves the fastest processing speed at 92.6 seconds per sample. This represents a dramatic improvement over existing baselines, outperforming

Mem0 by approximately $14\times$ (1350.9s) and A-Mem by over $50\times$ (5140.5s). This massive speedup is directly attributable to our semantic structured compression, which processes data in a streamlined single pass, thereby avoiding the complex graph updates required by Mem0 or the iterative summarization overheads inherent to A-Mem.

Beyond construction, SimpleMem also maintains the lowest retrieval latency at 388.3 seconds per sample, which is approximately 33% faster than LightMem and Mem0. This gain arises from the *adaptive retrieval* mechanism, which dynamically limits retrieval scope and prioritizes high-level abstract representations before accessing fine-grained details. By restricting retrieval to only the most relevant memory entries, the system avoids the expensive neighbor traversal and expansion operations that commonly dominate the latency of graph-based memory systems.

When considering the total time-to-insight, SimpleMem achieves a $4\times$ speedup over Mem0 and a $12\times$ speedup over A-Mem. Crucially, this efficiency does not come at the expense of performance. On the contrary, SimpleMem achieves the highest Average F1 among all compared methods. These results support our central claim that structured semantic compression and adaptive retrieval produce a more compact and effective reasoning substrate than raw context retention or graph-centric memory designs, enabling a superior balance between accuracy and computational efficiency.

*Table 4.* Comparison of construction time, retrieval time, total experiment time, and average F1 score across different memory systems (tested on LoCoMo-10 with GPT-4.1-mini; time values are reported as per-sample averages on LoCoMo-10).

| Model | Construction Time | Retrieval Time | Total Time | Average F1 |
|---|---|---|---|---|
| A-mem | 5140.5s | 796.7s | 5937.2s | 32.58 |
| Lightmem | 97.8s | 577.1s | 675.9s | 24.63 |
| Mem0 | 1350.9s | 583.4s | 1934.3s | 34.20 |
| **SimpleMem** | **92.6s** | **388.3s** | **480.9s** | **43.24** |

## 3.4. Ablation Study

In addition, we conduct an ablation study using the GPT-4.1-mini backend. We investigate the contribution of three key components. The results are summarized in Table 5.

**Impact of Semantic Structured Compression.** Replacing the proposed compression pipeline with standard chunk-based storage leads to a substantial degradation in temporal reasoning performance. Specifically, removing semantic structured compression reduces the Temporal F1 by 56.7%, from 58.62 to 25.40. This drop indicates that without context normalization steps such as resolving coreferences and converting relative temporal expressions into absolute timestamps, the retriever struggles to disambiguate events along the timeline. As a result, performance regresses to levels comparable to conventional retrieval-augmented generation systems that rely on raw or weakly structured context.

**Impact of Online Semantic Synthesis.** Disabling online semantic synthesis results in a 31.3% decrease in multi-hop reasoning performance. Without on-the-fly consolidation during the write phase, semantically related facts accumulate as fragmented entries, forcing the retriever to assemble dispersed evidence at query time. This fragmentation inflates contextual redundancy and rapidly exhausts the available context window in complex queries. The observed degradation demonstrates that proactive, intra-session synthesis is essential for maintaining a compact and semantically coherent memory topology, and for transforming local observations into reusable, high-density abstractions.

**Intent-Aware Retrieval Planning.** Removing intent-aware retrieval planning and reverting to a fixed-depth retrieval strategy primarily degrades performance on open-domain and single-hop tasks, with drops of 26.6% and 19.4%, respectively. In the absence of query-aware adjustment, the system either retrieves insufficient context for entity-specific queries or introduces excessive irrelevant information for simple queries. These results highlight the importance of dynamically modulating retrieval scope to balance relevance and efficiency during inference.

## 3.5. Case Study: Long-Term Temporal Grounding

To illustrate how SimpleMem handles long-horizon conversational history, Figure 3 presents a representative multi-session example spanning two weeks and approximately 24,000 raw tokens. SimpleMem filters low-information dialogue during ingestion and retains only high-utility memory entries, reducing the stored memory to about 800 tokens without losing task-relevant content.

**Temporal Normalization.** Relative temporal expressions such as last week" and yesterday" refer to different absolute times across sessions. SimpleMem resolves it into absolute timestamps at memory construction time, ensuring consistent temporal grounding over long interaction gaps.

**Precise Retrieval.** When queried about Sarah's past artworks, the intent-aware retrieval planner infers both the semantic focus (art-related activities) and the temporal constraints implied by the query. The system then performs parallel multi-view retrieval, combining semantic similarity with symbolic filtering to exclude unrelated activities and return only temporally valid entries. This example demonstrates how structured compression, temporal normalization, and adaptive retrieval jointly enable reliable long-term reasoning under extended interaction histories.

## 4. Related Work

**Memory Systems for LLM Agents.** Recent approaches manage memory through virtual context or structured representations. Virtual context methods, including MEMGPT (Packer et al., 2023), MEMORYOS (Kang et al., 2025), and

*Table 5.* Full ablation analysis with GPT-4.1-mini backend. The "Diff" columns indicate the percentage drop relative to the full SimpleMem model. The results confirm that each stage contributes significantly to specific reasoning capabilities.

| Configuration | Multi-hop | | Temporal | | Open Domain | | Single Hop | | Average | |
|---|---|---|---|---|---|---|---|---|---|---|
| | F1 | Diff | F1 | Diff | F1 | Diff | F1 | Diff | F1 | Diff |
| **Full SimpleMem** | **43.46** | - | **58.62** | - | **19.76** | - | **51.12** | - | **43.24** | - |
| w/o Semantic Compression | 34.20 | (↓21.3%) | 25.40 | (↓56.7%) | 17.50 | (↓11.4%) | 48.05 | (↓6.0%) | 31.29 | (↓27.6%) |
| w/o Online Synthesis | 29.85 | (↓31.3%) | 55.10 | (↓6.0%) | 18.20 | (↓7.9%) | 49.80 | (↓2.6%) | 38.24 | (↓11.6%) |
| w/o Intent-Aware Retrieval | 38.60 | (↓11.2%) | 56.80 | (↓3.1%) | 14.50 | (↓26.6%) | 41.20 | (↓19.4%) | 37.78 | (↓12.6%) |

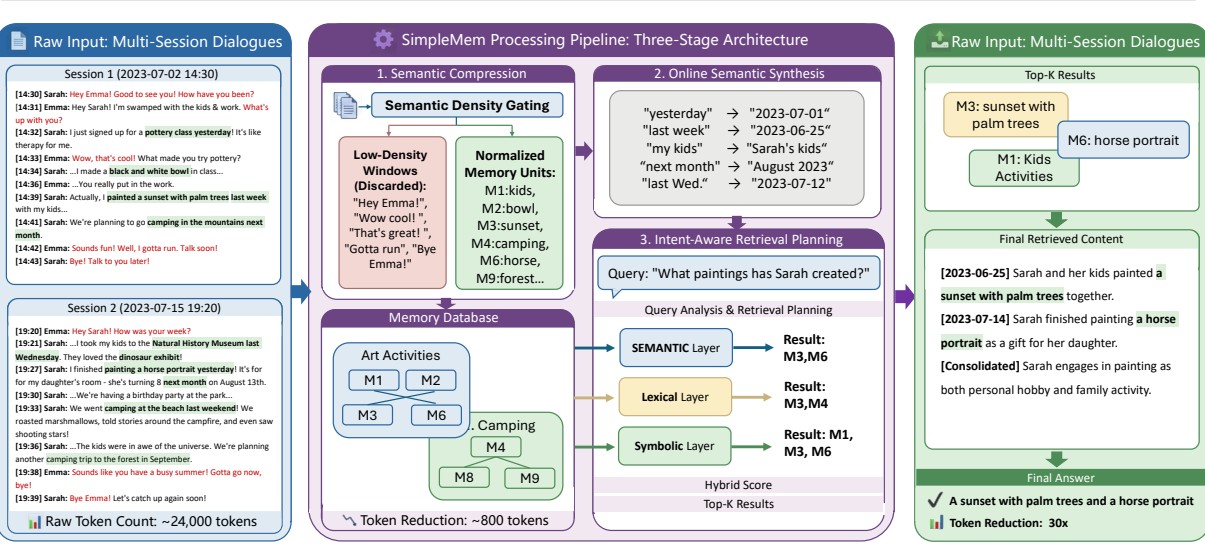

*Figure 3.* A Case of SimpleMem for Long-Term Multi-Session Dialogues. SimpleMem processes multi-session dialogues by filtering redundant content, normalizing temporal references, and organizing memories into compact representations. During retrieval, it adaptively combines semantic, lexical, and symbolic signals to select relevant entries.

SCM (Wang et al., 2023), extend interaction length via paging or stream-based controllers (Wang et al., 2024b) but typically store raw conversation logs, leading to redundancy and increasing processing costs. In parallel, structured and graph-based systems, such as MEMORYBANK (Zhong et al., 2024), MEM0 (Dev & Taranjeet, 2024), ZEP (Rasmussen et al., 2025), A-MEM (Xu et al., 2025), and O-MEM (Wang et al., 2025), impose structural priors to improve coherence but still rely on raw or minimally processed text, preserving referential and temporal ambiguities that degrade long-term retrieval. In contrast, SimpleMem adopts a semantic compression mechanism that converts dialogue into independent, self-contained facts, explicitly resolving referential and temporal ambiguities prior to storage.

**Context Management and Retrieval Efficiency.** Beyond memory storage, efficient access to historical information remains a core challenge. Existing approaches primarily rely on either long-context models or retrieval-augmented generation (RAG). Although recent LLMs support extended context windows (OpenAI, 2025; Deepmind, 2025; Anthropic, 2025), and prompt compression methods aim to reduce costs (Jiang et al., 2023a; Liskavetsky et al., 2025), empirical studies reveal the "Lost-in-the-Middle" effect (Liu et al., 2023; Kuratov et al., 2024), where reasoning performance degrades as context length increases, along-

side prohibitive computational overhead for lifelong agents. RAG-based methods (Lewis et al., 2020; Asai et al., 2023; Jiang et al., 2023b), including structurally enhanced variants such as GRAPHRAG (Edge et al., 2024; Zhao et al., 2025) and LIGHTRAG (Guo et al., 2024), decouple memory from inference but are largely optimized for static knowledge bases, limiting their effectiveness for dynamic, time-sensitive episodic memory. In contrast, SimpleMem improves retrieval efficiency through *Intent-Aware Retrieval Planning*, jointly leveraging semantic, lexical, and symbolic signals to construct query-specific retrieval plans and dynamically adapt the retrieval budget, achieving token-efficient reasoning under constrained context budgets.

## 5. Conclusion

We introduce SimpleMem, an efficient agent memory architecture grounded in the principle of semantic lossless compression. By treating memory as an active process rather than passive storage, SimpleMem integrates *Semantic Structured Compression* to filter noise at the source, *Online Semantic Synthesis* to consolidate fragmented observations during writing, and *Intent-Aware Retrieval Planning* to dynamically adapt retrieval scope. Empirical evaluation on the LoCoMo and LongMemEval-S benchmark demonstrates the effectiveness and efficiency of our method.

## Impact Statement

This work aims to advance the field of machine learning. While it may have broader societal implications, we do not identify any specific impacts that require separate discussion at this time.

## Acknowledgement

This work is partially supported by Amazon Research Award, Cisco Faculty Research Award, and Coefficient Giving. This work is also partly supported by the National Center for Transportation Cybersecurity and Resiliency (TraCR) (a U.S. Department of Transportation National University Transportation Center) headquartered at Clemson University, Clemson, South Carolina, USA (USDOT Grant #69A3552344812). Any opinions, findings, conclusions, and recommendations expressed in this material are those of the author(s) and do not necessarily reflect the views of TraCR, and the U.S. Government assumes no liability for the contents or use thereof.

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

# A. Detailed System Prompts

To ensure full reproducibility of the **SimpleMem** pipeline, we provide the exact system prompts used in the key processing stages. All prompts are designed to be model-agnostic but were optimized for GPT-4o-mini in our experiments to ensure cognitive economy.

## A.1. Stage 1: Semantic Structured Compression Prompt

This prompt performs entropy-aware filtering and context normalization. Its goal is to transform raw dialogue windows into compact, context-independent memory units while excluding low-information interaction content.

*Listing 1.* Prompt for Semantic Structured Compression and Normalization.

```
You are a memory encoder in a long-term memory system. Your task is to transform raw
    conversational input into compact, self-contained memory units.

INPUT METADATA:
Window Start Time: {window_start_time} (ISO 8601)
Participants: {speakers_list}

INSTRUCTIONS:
1. Information Filtering:
   - Discard social filler, acknowledgements, and conversational routines that introduce
       no new factual or semantic information.
   - Discard redundant confirmations unless they modify or finalize a decision.
   - If no informative content is present, output an empty list.

2. Context Normalization:
   - Resolve all pronouns and implicit references into explicit entity names.
   - Ensure each memory unit is interpretable without access to prior dialogue.

3. Temporal Normalization:
   - Convert relative temporal expressions (e.g., "tomorrow", "last week") into absolute
       ISO 8601 timestamps using the window start time.

4. Memory Unit Extraction:
   - Decompose complex utterances into minimal, indivisible factual statements.

INPUT DIALOGUE:
{dialogue_window}

OUTPUT FORMAT (JSON):
{
  "memory_units": [
    {
      "content": "Alice agreed to meet Bob at the Starbucks on 5th Avenue on 2025-11-20T14
          :00:00.",
      "entities": ["Alice", "Bob", "Starbucks", "5th Avenue"],
      "topic": "Meeting Planning",
      "timestamp": "2025-11-20T14:00:00",
      "salience": "high"
    }
  ]
}
```

## A.2. Stage 2: Adaptive Retrieval Planning Prompt

This prompt analyzes the user query prior to retrieval. Its purpose is to estimate query complexity and generate a structured retrieval plan that adapts retrieval scope accordingly.

*Listing 2.* Prompt for Query Analysis and Adaptive Retrieval Planning.

```
Analyze the following user query and generate a retrieval plan. Your objective is to
    retrieve sufficient information while minimizing unnecessary context usage.
```

```
USER QUERY:
{user_query}

INSTRUCTIONS:
1. Query Complexity Estimation:
   - Assign "LOW" if the query can be answered via direct fact lookup or a single memory
      unit.
   - Assign "HIGH" if the query requires aggregation across multiple events, temporal
      comparison, or synthesis of patterns.

2. Retrieval Signals:
   - Lexical layer: extract exact keywords or entity names.
   - Temporal layer: infer absolute time ranges if relevant.
   - Semantic layer: rewrite the query into a declarative form suitable for semantic
      matching.

OUTPUT FORMAT (JSON):
{
  "complexity": "HIGH",
  "retrieval_rationale": "The query requires reasoning over multiple temporally separated
      events.",
  "lexical_keywords": ["Starbucks", "Bob"],
  "temporal_constraints": {
    "start": "2025-11-01T00:00:00",
    "end": "2025-11-30T23:59:59"
  },
  "semantic_query": "The user is asking about the scheduled meeting with Bob, including
      location and time."
}
```

### A.3. Stage 3: Reconstructive Synthesis Prompt

This prompt guides the final answer generation using retrieved memory. It combines high-level abstract representations with fine-grained factual details to produce a grounded response.

*Listing 3.* Prompt for Reconstructive Synthesis (Answer Generation).

```
You are an assistant with access to a structured long-term memory.

USER QUERY:
{user_query}

RETRIEVED MEMORY (Ordered by Relevance):

[ABSTRACT REPRESENTATIONS]:
{retrieved_abstracts}

[DETAILED MEMORY UNITS]:
{retrieved_units}

INSTRUCTIONS:
1. Hierarchical Reasoning:
   - Use abstract representations to capture recurring patterns or general user
      preferences.
   - Use detailed memory units to ground the response with specific facts.

2. Conflict Handling:
   - If inconsistencies arise, prioritize the most recent memory unit.
   - Optionally reference abstract patterns when relevant.

3. Temporal Consistency:
   - Ensure all statements respect the timestamps provided in memory.
```

```
4. Faithfulness:
   - Base the answer strictly on the retrieved memory.
   - If required information is missing, respond with: "I do not have enough information
      in my memory."

FINAL ANSWER:
```

### A.4. LongMemEval Evaluation Prompt

For the LongMemEval benchmark, we employed `gpt-4.1-mini` as the judge to evaluate the correctness of the agent's responses. The prompt strictly instructs the judge to focus on semantic and temporal consistency rather than exact string matching. The specific prompt template used is provided below:

*Listing 4.* LLM-as-a-Judge Evaluation Prompt.

```
Your task is to label an answer to a question as 'CORRECT' or 'WRONG'.
You will be given the following data:
    (1) a question (posed by one user to another user),
    (2) a 'gold' (ground truth) answer,
    (3) a generated answer
which you will score as CORRECT/WRONG.

The point of the question is to ask about something one user should know about the other
    user based on their prior conversations.
The gold answer will usually be a concise and short answer that includes the referenced
    topic, for example:
Question: Do you remember what I got the last time I went to Hawaii?
Gold answer: A shell necklace

The generated answer might be much longer, but you should be generous with your grading -
    as long as it touches on the same topic as the gold answer, it should be counted as
    CORRECT.

For time related questions, the gold answer will be a specific date, month, year, etc. The
     generated answer might be much longer or use relative time references (like "last
    Tuesday" or "next month"), but you should be generous with your grading - as long as
    it refers to the same date or time period as the gold answer, it should be counted as
    CORRECT. Even if the format differs (e.g., "May 7th" vs "7 May"), consider it CORRECT
    if it's the same date.

Now it's time for the real question:
Question:  {question}
Gold answer: {gold_answer}
Generated answer: {generated_answer}

First, provide a short (one sentence) explanation of your reasoning, then finish with
    CORRECT or WRONG.
Do NOT include both CORRECT and WRONG in your response, or it will break the evaluation
    script.

Just return the label CORRECT or WRONG in a json format with the key as "label".
```

## B. Extended Implementation Details and Experiments

### B.1. Dataset Description

**LoCoMo** (Maharana et al., 2024) is specifically designed to test the limits of LLMs in processing long-term conversational dependencies. The dataset comprises conversation samples ranging from 200 to 400 turns, containing complex temporal shifts and interleaved topics. The evaluation set consists of 1,986 questions categorized into four distinct reasoning types: (1) Multi-Hop Reasoning: Questions requiring the synthesis of information from multiple disjoint turns (e.g., `"Based on what X said last week and Y said today..."`); (2) Temporal Reasoning: Questions testing the model's

ability to understand event sequencing and absolute timelines (e.g., "Did X happen before Y?"); (3) Open Domain: General knowledge questions grounded in the conversation context; (4) Single Hop: Direct retrieval tasks requiring exact matching of specific facts.

**LongMemEval-S** benchmark. The defining characteristic of this dataset is its *extreme context length*, which poses a unique and severe challenge for memory systems. Unlike standard benchmarks, LongMemEval-S requires the system to precisely locate specific answers across various sub-categories (e.g., temporal events, user preferences) within an exceptionally long interaction history. This massive search space significantly escalates the difficulty of retrieval and localization, serving as a rigorous stress test for the system's precision. We utilized an *LLM-as-a-judge* protocol (using gpt-4.1-mini) to score the correctness of generated answers against ground-truth references, categorizing responses as either CORRECT or WRONG based on semantic and temporal alignment. The full evaluation prompt is provided in Appendix A.4.

### B.2. Hyperparameter Configuration

Table 7 summarizes the hyperparameters used to obtain the results reported in Section 3. These values were selected to balance memory compactness and retrieval recall, with particular attention to the thresholds governing semantic structured compression and recursive consolidation.

### B.3. Hyperparameter Sensitivity Analysis

To assess the effectiveness of semantic structured compression and to motivate the design of adaptive retrieval, we analyze system sensitivity to the number of retrieved memory entries ($k$). We vary $k$ from 1 to 20 and report the average F1 score on the LoCoMo benchmark using the GPT-4.1-mini backend.

Table 6 provides two key observations. First, rapid performance saturation is observed at low retrieval depth. SimpleMem achieves strong performance with a single retrieved entry (35.20 F1) and reaches approximately 99% of its peak performance at $k = 3$. This behavior indicates that semantic structured compression produces memory units with high information content, often sufficient to answer a query without aggregating many fragments.

Second, robustness to increased retrieval depth distinguishes SimpleMem from baseline methods. While approaches such as MemGPT experience performance degradation at larger $k$, SimpleMem maintains stable accuracy even when retrieving up to 20 entries. This robustness enables adaptive retrieval to safely expand context for complex reasoning tasks without introducing excessive irrelevant information.

*Table 6.* Performance sensitivity to retrieval count ($k$). **SimpleMem** demonstrates "Rapid Saturation," reaching near-optimal performance at $k = 3$ (42.85) compared to its peak at $k = 10$ (43.45). This validates the high information density of Atomic Entries, proving that huge context windows are often unnecessary for accuracy.

| Method | Top-$k$ Retrieved Entries | | | | |
| --- | --- | --- | --- | --- | --- |
| | $k$=1 | $k$=3 | $k$=5 | $k$=10 | $k$=20 |
| ReadAgent | 6.12 | 8.45 | 9.18 | 8.92 | 8.50 |
| MemGPT | 18.40 | 22.15 | 25.59 | 24.80 | 23.10 |
| **SimpleMem** | **35.20** | **42.85** | **43.24** | **43.45** | **43.40** |

*Table 7.* Detailed hyperparameter configuration for SimpleMem. The system employs adaptive thresholds to balance memory compactness and retrieval effectiveness.

| Module | Parameter | Value / Description |
| --- | --- | --- |
| *Stage 1: Semantic Structured Compression* | Window Size ($W$) | 20 turns |
| | Sliding Stride | 5 turns (25% overlap) |
| | Model Backend | gpt-4.1-mini (temperature = 0.0) |
| | Output Constraint | Strict JSON schema enforced |
| *Stage 2: Online Semantic Synthesis* | Embedding Model | Qwen3-embedding-0.6b (1024 dimensions) |
| | Vector Database | LanceDB (v0.4.5) with IVF-PQ indexing |
| | Stored Metadata | timestamp, entities, topic, salience |
| *Stage 3: Intent-Aware Retrieval Planning* | Query Complexity Estimator | gpt-4.1-mini |
| | Retrieval Range | [3, 20] |
| | Minimum Depth | 1 |
| | Maximum Depth | 20 |
| | Re-ranking | Disabled (multi-view score fusion applied directly) |

