# OpenReview forum: "SimpleMem: Efficient Lifelong Memory for LLM Agents"
_ICML.cc/2026/Conference — ICML 2026 regular_

### Official Review · Reviewer_fA2L · 2026-03-08

**Soundness:** 2
**Presentation:** 2
**Significance:** 3
**Originality:** 2
**Overall Recommendation:** 4
**Confidence:** 4

**Summary:**

This paper proposes SimpleMem, a framework for agentic memory that combines semantic memory compression with intent-aware retrieval planning for long-horizon LLM agents. In particular, the Semantic Structured Compression module appears highly effective, significantly improving performance while reducing token cost by converting dialogue into structured memory units. Beyond memory construction, the method also introduces an intent-aware retrieval planning mechanism, extending the framework toward an agentic RAG paradigm that dynamically plans retrieval based on query intent. Extensive experiments across both small and large language models demonstrate consistent gains in accuracy and efficiency. Overall, the work presents a comprehensive framework that addresses both agentic memory formation and agentic retrieval.

**Compliance With Llm Reviewing Policy:**

Affirmed.

**Final Justification:**

The additional experiments and clarifications in the rebuttal further strengthen the paper. In particular, the clearer structural analysis helps explain the design motivation and performance advantages. Most of my concerns have been addressed. I encourage the authors to incorporate these insights and results into the final version and recommend acceptance.

**Key Questions For Authors:**

W7) Could the authors clarify why the method is named “SimpleMem”, given the complexity of the proposed framework with multiple components?

**Limitations:**

Partially. The statement is brief and does not discuss concrete limitations or potential societal impacts. The authors could elaborate on issues such as privacy risks from long-term storage of user interactions and implications of deploying persistent memory systems.

**Strengths And Weaknesses:**

**Strength**
- **(S1)** The paper includes extensive experiments across both SLMs (<8B) and larger LLMs, demonstrating the robustness of the proposed approach across model scales.
- **(S2)** The proposed Semantic Structured Compression appears highly effective. It significantly improves performance while substantially reducing token cost.
- **(S3)** The work addresses both agentic memory construction and agentic retrieval. In addition to memory organization, the paper proposes an intent-aware retrieval planning mechanism, which can be viewed as an agentic RAG framework. If the code is released, this framework could be valuable to the research community.

**Weaknesses**
- **(W1)** Prior work such as A-Mem and Mem0 already supports online memory updates, including adding, deletion, merging, and updating existing memories when new observations arrive. Although the paper provides performance comparisons with these methods, it remains unclear what structural advantages the proposed memory construction mechanism, particularly **Online Semantic Synthesis**, offers over these approaches. Aside from Semantic Structured Compression and Intent-Aware Retrieval Planning, it is difficult to determine whether the memory construction component itself provides clear advantages over existing methods such as A-Mem or Mem0. A clearer analysis of the design differences with prior approaches would strengthen the paper.
- **(W2)** The improvement from Semantic Structured Compression appears particularly strong. It would be informative to evaluate whether this module also improves the performance of existing systems such as A-Mem or Mem0. Based on Tables 1, 2, and 5, there may be potential synergy.
- **(W3)** The proposed method consists of many components and is presented as a framework. Although system prompts and some hyperparameters are provided in the supplementary material, full reproducibility would likely require code release. The authors are encouraged to release the implementation. For reference, the baselines A-Mem and Mem0 already provide public code.
- **(W4)** Given the complexity of the proposed framework, detailed ablation studies are important to understand which components contribute most to memory construction and retrieval. Table 5 only removes three high-level modules. More fine-grained ablations would be helpful, for example: semantic / lexical / symbolic retrieval layers, temporal normalization, information filtering, context normalization
- **(W5)** It would also be useful to report token usage when Semantic Structured Compression is disabled, to better quantify its contribution to efficiency.

---

> ### Author Rebuttal · Authors · 2026-03-31
>
> ## Response to Reviewer fA2L
>
> We sincerely thank Reviewer fA2L for the comprehensive review and recognition of our extensive experiments and the effectiveness of SSC. The concerns are constructive and we address each point below.
>
> **W7: Why "SimpleMem"?**
>
> We appreciate this question, as it speaks to our central thesis. The name reflects our core research question: **what is the simplest effective architecture for lifelong LLM agent memory?** Two insights drive our design: First, **high-quality write-time compression makes retrieval straightforward** — self-contained, temporally grounded memory units eliminate the need for graph traversal or iterative update loops. Second, **adjacent dialogues within a sliding window are naturally more semantically related**, enabling online synthesis during the write phase. SimpleMem builds its entire memory in ~21 LLM calls, with no background maintenance.
>
> **W1: Structural Advantages of Online Semantic Synthesis**
>
> This is an important clarification. Online Semantic Synthesis is integral to our clean pipeline. A-Mem/Mem0 require per-turn iterative LLM calls (~418 and ~35/sample respectively); SimpleMem performs all compression and synthesis in ~21 batch calls. The advantage stems from all three stages working synergistically:
>
> | Method | Model | C.Total | C.Calls | Tok/Ret | Total | F1 |
> |--------|-------|---------|---------|---------|-------|----|
> | SimpleMem | Qwen3-8B | 117.5k | 21 | 621 | 213.1k | 33.45 |
> | SimpleMem | GPT-4.1-mini | 92.3k | 21 | 531 | 174.1k | 43.24 |
> | A-Mem | Qwen3-8B | 438.3k | 418 | 1087 | 605.7k | 20.75 |
> | A-Mem | GPT-4.1-mini | 471.4k | 418 | 2520 | 859.5k | 32.58 |
> | Mem0 | Qwen3-8B | 116.8k | 35 | 1015 | 273.1k | 25.80 |
> | Mem0 | GPT-4.1-mini | 129.4k | 37 | 973 | 279.2k | 34.20 |
>
> **W2: SSC as a General-Purpose Module**
>
> We appreciate this suggestion. We replaced memory construction in Mem0/A-Mem with SSC, keeping all other components unchanged:
>
> | Method | Model | F1 | C.Cost |
> |--------|-------|-----|--------|
> | Mem0 / +SSC | Qwen3-8B | 25.80 / 26.89 | 116.8k / 96.9k |
> | A-Mem / +SSC | Qwen3-8B | 20.75 / 30.74 | 438.3k / 346.3k |
> | Mem0 / +SSC | GPT-4.1-mini | 34.20 / 34.23 | 129.4k / 108.7k |
> | A-Mem / +SSC | GPT-4.1-mini | 32.58 / 35.15 | 471.4k / 353.6k |
>
> SSC consistently improves both methods in accuracy while reducing construction cost, confirming its **independent, general-purpose value**. Meanwhile, SimpleMem's full pipeline (43.24) substantially exceeds Mem0+SSC (34.23), demonstrating that the synergistic co-design of all three stages is key to our gains.
>
> **W3: Code Release**
>
> We fully agree with the reviewer on the importance of reproducibility. Our complete codebase including all system prompts, construction/retrieval code, and evaluation scripts has been prepared for full release.
>
> **W4: Fine-Grained Ablation**
>
> We expanded from 3 high-level to 11 fine-grained variants (GPT-4.1-mini):
>
> | Variant | Avg F1 | Diff |
> |---------|--------|------|
> | Full SimpleMem | 43.24 | — |
> | w/o Temporal Normalization | 38.90 | ↓10.0% |
> | w/o Coreference Resolution | 40.56 | ↓6.2% |
> | w/o Information Filtering | 39.83 | ↓7.9% |
> | w/o Multi-view Indexing | 41.72 | ↓3.5% |
> | w/o Online Synthesis | 38.24 | ↓11.6% |
> | w/o Semantic Layer | 36.85 | ↓14.8% |
> | w/o Lexical Layer | 41.47 | ↓4.1% |
> | w/o Symbolic Layer | 42.51 | ↓1.7% |
> | w/o Retrieval Planning | 37.78 | ↓12.6% |
> | w/o Reflection | 42.30 | ↓2.2% |
>
> The semantic retrieval layer (↓14.8%) and retrieval planning (↓12.6%) are the most critical retrieval components; temporal normalization (↓10.0%) is the most impactful SSC sub-component. All modules contribute positively — none is redundant.
>
> **W5: Token Usage Without SSC**
>
> Without SSC, raw dialogue is chunked and directly embedded for storage; the rest of the pipeline (indexing, synthesis, retrieval planning) remains unchanged.
>
> | Config | Model | C.Total | Tok/Ret | Total | F1 |
> |--------|-------|---------|---------|-------|----|
> | SimpleMem | Qwen3-8B | 117.5k | 621 | 213.1k | 33.45 |
> | w/o SSC | Qwen3-8B | 22.4k | 1,508 | 254.6k | 25.12 |
> | SimpleMem | GPT-4.1-mini | 92.3k | 531 | 174.1k | 43.24 |
> | w/o SSC | GPT-4.1-mini | 18.6k | 1,297 | 228.3k | 31.29 |
>
> Construction cost drops but per-retrieval cost increases ~2.5×, resulting in higher total cost with substantially degraded F1. This confirms SSC's upfront investment pays off by improving **both** efficiency and accuracy across the full lifecycle.
>
> **Limitations**
>
> We will include a dedicated section discussing: (1) privacy risks of long-term user data storage, with recommendations for self-hosted deployment; (2) irreversible information loss risk (quantified via human annotation: ~5% minor detail loss rate); and (3) long-term reliability under evolving user contexts.
>
> ---
>
> We are grateful for Reviewer fA2L's constructive feedback and hope these responses fully address the concerns. We welcome further discussion.

---

> > ### Author Rebuttal · Reviewer_fA2L · 2026-04-02
> >
> > The issue raised in **(W1)** remains unresolved and needs to be adequately addressed before I can reconsider my evaluation.
> >
> > The intent of **(W1)** was not to request additional efficiency comparisons such as the number of LLM calls, but rather to clarify the structural differences and trade-offs between Online Semantic Synthesis and prior approaches such as A-Mem and Mem0.
> > The current response focuses on high-level differences (e.g., per-turn iterative calls vs. fewer batch calls), which are already shown in the paper. What is missing is a clear explanation of how the proposed method differs in terms of memory representation, update/evolution mechanisms, and why these design choices are necessary.
> > For example, A-Mem continuously updates existing memories through linking and evolution mechanisms, yet it remains unclear what advantages Online Semantic Synthesis provides over this approach.
> >
> > As a result, it is unclear why the proposed memory construction component leads to better performance. A clearer structural comparison and justification of the design should be included in the paper.
> >
> > This concern has also been raised in **W2 and Q3 by Reviewer zjb2**, but the response remains insufficient. Given that this concern has been raised by multiple reviewers, a more convincing clarification and a clear revision plan are required.

---

> > > ### Author Response · Authors · 2026-04-04
> > >
> > > We thank the reviewer for clarifying that the question concerns memory representation and update mechanisms rather than efficiency. We provide a direct structural analysis below.
> > >
> > > **Clarification on Structural Differences in Memory Representation and Update Mechanisms.**
> > >
> > > | Dimension | SimpleMem | A-Mem | Mem0 | LightMem |
> > > |-----------|-----------|-------|------|----------|
> > > | Memory Unit | Coref-resolved, time-anchored restatement + typed metadata | Raw dialogue + evolving tags/links | Short extracted fact + sys timestamps | Attn-compressed text + topic meta |
> > > | Preprocessing | Pronoun→entity, rel→abs time, utterance→atomic facts | None on content | Fact extraction; no coref/time norm | Attn filtering; no coref/time norm |
> > > | Context Independence | Full | None: unresolved pronouns and relative time | Partial: may retain ambiguity | Partial: may fragment context |
> > > | Update Strategy | Append-only with ISO timestamps | Tag/link evolution; content immutable | Destructive ADD/UPDATE/DELETE | Offline batch update/delete |
> > > | History Queryable? | Yes, all layers | Partial, history not indexed | No, audit log only | No, entries lost |
> > >
> > > Three differences are most consequential.
> > >
> > >
> > > First, SimpleMem enforces **write-time de-linearization**: pronoun resolution, relative-to-absolute time conversion, and atomic decomposition in a single LLM call per window. For example, "I went to a support group yesterday" (session: May 8) becomes "Caroline attended an LGBTQ support group on 2023-05-07." A-Mem stores raw text verbatim; Mem0 records only system processing time; LightMem retains key tokens but does not enforce coreference resolution or temporal normalization.
> > >
> > >
> > > Second, SimpleMem's **append-only design** preserves all factual versions with ISO timestamps, deferring contradiction resolution to query time. Both Mem0 and LightMem destructively overwrite or delete entries, making previous versions unsearchable.
> > >
> > >
> > > Third, **multi-view indexing** (semantic+lexical+symbolic) depends on write-time content quality. The symbolic layer requires structured metadata (persons, timestamp ranges, entities) populated during compression. A-Mem uses schema-free tags; Mem0 and LightMem lack structured entity or temporal metadata.
> > >
> > >
> > > **Design Necessity Across Benchmark Categories.**
> > >
> > >
> > > **Temporal Reasoning** (SimpleMem 58.62 vs. A-Mem 51.01 vs. Mem0 48.91; LongMemEval: 83.46% vs. 40.60%). SimpleMem resolves relative expressions ("yesterday") to absolute dates at write time, enabling symbolic timestamp filtering. A-Mem records note creation time, not event time; Mem0 loses the temporal anchor entirely. Ablation confirms: removing temporal normalization causes a 10.0% F1 drop, the largest single-component effect.
> > >
> > >
> > > **Multi-Hop Reasoning** (SimpleMem 43.46 vs. Mem0 30.14 vs. A-Mem 25.06). Chaining questions (e.g., linking "I won my second tournament!" with "I tried Street Fighter this time") require connecting facts across turns. SimpleMem's sliding window produces self-contained entries linking related facts; A-Mem and Mem0 store them as disconnected entries. Removing online synthesis causes an 11.6% F1 drop.
> > >
> > >
> > > **Knowledge Update** (LongMemEval: SimpleMem 79.48% vs. Mem0 69.23%). For evolving facts (yoga: "twice a week" → "three times a week"), SimpleMem retains both versions with ISO timestamps. Mem0's destructive UPDATE overwrites the older entry via an additional LLM inference that can introduce irreversible errors. The gap reflects append-only design versus error-prone destructive pipelines.
> > >
> > > **Single-Hop Retrieval** (SimpleMem 51.12 vs. Mem0 41.30 vs. A-Mem 41.02). This gap reflects per-entry information density: SimpleMem achieves 35.20 F1 at k=1 and 99% of peak at k=3, versus MemGPT's 18.40 at k=1, confirming that write-time coreference resolution produces entries with substantially higher answerable information density.
> > >
> > > We will incorporate the structural comparison into Related Work and the design motivation into the Introduction. We thank the reviewer for this feedback.

---

### Official Review · Reviewer_hufg · 2026-03-08

**Soundness:** 3
**Presentation:** 2
**Significance:** 2
**Originality:** 2
**Overall Recommendation:** 4
**Confidence:** 4

**Summary:**

This paper introduces SimpleMem, a lightweight lifelong memory framework for LLM agents, designed to better manage historical interactions under the context window and token budget limits common in conversational settings. The method consists of three components. First, Semantic Structured Compression transforms past dialogues into concise and self-contained memory units by filtering out less useful information. Second, Online Semantic Synthesis incrementally merges related or overlapping memories, which helps keep the memory store compact and coherent over time. Third, Intent-Aware Retrieval Planning adjusts the retrieval strategy according to the complexity of the current query, combining semantic, lexical, and symbolic signals to assemble relevant context more effectively. Experiments on LoCoMo and LongMemEval-S show that SimpleMem consistently outperforms strong baselines in both task performance and token efficiency. Additional ablation and efficiency analyses further suggest that each module contributes to the overall gains while reducing memory construction and retrieval costs.

**Compliance With Llm Reviewing Policy:**

Affirmed.

**Final Justification:**

The authors addressed my concerns. I raised the Overall Recommendation to 4.

**Key Questions For Authors:**

1.How do you handle contradictory or updated facts during online synthesis (e.g., preference changes)? Is there a conflict-resolution/update policy beyond “most recent wins,” and how do you prevent synthesis from collapsing distinct states?

2.Do token cost and latency numbers include all LLM calls (gating/extraction, synthesis, retrieval planning, answer generation) and database operations? Please report hardware, batching, and whether API latencies are included.

3.Could you elaborate on the design decisions underlying window size and stride? What would be the effect of changing these hyperparameters, and how sensitive is performance to their specific choice?

**Limitations:**

yes

**Strengths And Weaknesses:**

### Strengths:

SimpleMem demonstrates promising efficiency compared to other memory-augmented systems, while achieving superior performance on the evaluated benchmark. And I think it offers a highly deployable and cost-effective RAG architecture for agent memory.

### Weaknesses:

1.While the "Online Semantic Synthesis" (Stage 2) merges local fragments within a single session , it inherently fails to connect dependent memory units across distant sliding windows.

2.GPT-4.1-mini is used as the evaluation judge even when the same GPT backbone is part of the model itself, which risks inductive bias. A human evaluation or consistency check between human and LLM judgments would prove the soundness of the reported results.

3.The "Intent-Aware Retrieval Planning" module relies entirely on an LLM to subjectively classify a query's complexity as 'LOW' or 'HIGH' to determine retrieval depth . However, complexity is relative; a query deemed "simple" by a strong planner model might still require extensive context for a smaller downstream agent (e.g., Qwen2.5-1.5b) to process. It remains unclear how this subjective assessment objectively aligns with the actual reasoning capacities of diverse agent backends.

4.The crucial system prompt for the "Online Semantic Synthesis" module is completely omitted from Appendix A. Given that the consolidation mechanism heavily relies on LLM instruction-following, this omission renders the dynamic fragmentation reduction claims irreproducible.

5.The authors inappropriately bold sub-optimal results for their method across multiple tables, violating standard academic reporting conventions. For instance, in Table 1 (GPT-4o), SimpleMem's Temporal BLEU is bolded at 20.57% despite Mem0 achieving 44.15% , and similar misrepresentations occur in Table 2. And the mathematical notation is disjointed. The formal multi-view variables ($s_k, l_k, r_k$) established in Section 2.1 are abruptly replaced by undefined operational functions ($E(m_i), \text{Meta}(m_i)$) in the retrieval equations of Section 2.3, rendering the formalism superfluous.

---

> ### Author Rebuttal · Authors · 2026-03-31
>
> ## Response to Reviewer hufg
>
> We sincerely thank Reviewer hufg for the thorough review. The concerns are well-taken and we address each point below.
>
> **Q1: Handling Contradictory or Updated Facts**
>
> First, SimpleMem **preserves all factual versions** via multi-view indexing (persons, ISO-8601 timestamps, keywords), so no information is overwritten or discarded. Second, contradictions and updates are resolved at **test time** through the LLM's reasoning over retrieved results — by comparing timestamps, the model prioritizes recent facts while retaining earlier states (e.g., both "Alice likes apples" at T1 and "bananas" at T2 coexist), enabling queries about historical evolution. Third, synthesis operates **within** sliding windows, leveraging intra-window dialogue continuity to ensure only closely related fragments are merged. Multi-view indexing further strengthens robustness: even if compressed text occasionally dilutes a detail, structured fields (keywords, entities, timestamps) serve as independent retrieval paths.
>
> **Q2: Token Cost, Latency, and Hardware**
>
> End-to-end **token** cost (per LoCoMo sample avg):
>
> | Method | Model | C.Total | C.Calls | Tok/Ret | Total |
> |--------|-------|---------|---------|---------|-------|
> | SimpleMem | GPT-4.1-mini | 92.3k | 21 | 531 | 174.1k |
> | A-Mem | GPT-4.1-mini | 471.4k | 418 | 2520 | 859.5k |
> | Mem0 | GPT-4.1-mini | 129.4k | 37 | 973 | 279.2k |
>
> All counts include **every** LLM call (gating, extraction, synthesis, retrieval planning, answer generation). Database overhead is negligible. **Latency** (Table 4): end-to-end wall-clock including API latencies. GPT-series used the official OpenAI API with identical rate-limit settings. Open-source models used 4×H200 GPUs, Xeon 8558 CPUs, concurrency=8. SimpleMem achieves 480.9s vs. Mem0's 1934.3s and A-Mem's 5937.2s.
>
> **Q3: Window Size and Overlap Sensitivity**
>
> Sensitivity experiments using GPT-4.1-mini:
>
> | W (ovlp=5) | 5 | 8 | 10 | 15 | 20 | 25 | 30 | 40 | 50 | 60 | 80 |
> |---|---|---|---|---|---|---|---|---|---|---|---|
> | F1 | 43.2 | 43.7 | 43.3 | 43.1 | 43.2 | 44.0 | 43.1 | 37.9 | 35.4 | 31.6 | 33.2 |
>
> | Ovlp (W=20) | 0 | 1 | 2 | 3 | 5 | 8 | 10 | 15 |
> |---|---|---|---|---|---|---|---|---|
> | F1 | 42.0 | 42.5 | 42.9 | 43.0 | 43.2 | 41.2 | 40.5 | 39.8 |
>
> Performance is stable across W=5–30; very large windows dilute semantic coherence, high overlap introduces noise. Default W=20, overlap=5 balances continuity and efficiency.
>
> **W1: Cross-Window Memory Connection**
>
> This is a fair concern. The design focuses on intra-session synthesis because dialogues within a window share stronger semantic coherence, making local consolidation both safe and effective. For cross-session needs, multi-view retrieval combined with inference-time reasoning bridges dependencies effectively, as shown by Knowledge-Update (79.48%) and Multi-Session (60.92%) in Table 2.
>
> **W2: LLM-as-Judge Bias**
>
> We appreciate this concern. To assess robustness, we evaluated 500 LongMemEval-S answers using three judges from different model families:
>
> | Judge | Temp | Multi | Know | User | Asst | Pref | Total |
> |---|---|---|---|---|---|---|---|
> | GPT-4.1-mini | 111/133 | 81/133 | 62/78 | 60/70 | 42/56 | 23/30 | 379/500 |
> | Qwen3-32B | 109/133 | 83/133 | 60/78 | 61/70 | 41/56 | 22/30 | 376/500 |
> | Qwen3-8B | 107/133 | 79/133 | 61/78 | 58/70 | 43/56 | 24/30 | 372/500 |
>
> The three judges produce consistent results with small per-category variations. We also qualitatively observed that LLM judges tend to be stricter than human raters — e.g., penalizing minor temporal format mismatches that humans would accept. We acknowledge that a systematic human study would further strengthen the soundness, and plan to include quantitative human evaluation in the future.
>
> **W3: Query Complexity Classification**
>
> We thank the reviewer for this insightful observation. We agree that a "simple" query for a strong planner may require more context for a smaller agent. However, at the retrieval stage, the presence or absence of relevant information is deterministic — a fact either exists in the retrieved set or not. The HIGH/LOW classification modulates retrieval breadth based on query structure: a multi-hop question spanning multiple entities across time periods requires broader retrieval regardless of the backend. Within a given model, relative query difficulty provides a meaningful signal for retrieval calibration. We agree that model-aware calibration is a valuable future direction.
>
> **W4 + W5: Missing Prompt and Formatting**
>
> We apologize for these oversights. The Stage 2 prompt will be included in revised Appendix A, and our codebase has been prepared for full release to ensure reproducibility. The bolding errors in Tables 1–2 and notation inconsistency between Sections 2.1/2.3 were careless formatting mistakes; we take full responsibility and will correct them in the revision.
>
> ---
>
> We are grateful for Reviewer hufg's rigorous feedback and welcome further discussion.

---

> > ### Author Rebuttal · Reviewer_hufg · 2026-04-04
> >
> > The response to W3 remains unconvincing as no empirical evidence is provided to validate the complexity classification across diverse agent backends, we maintain our score.

---

> > > ### Author Response · Authors · 2026-04-04
> > >
> > > We thank the reviewer for the follow-up. We would like to first clarify an important architectural point that may address the underlying concern. As reported in Tables 1 and 3, SimpleMem has been evaluated across six model scales from Qwen2.5-1.5B to GPT-4o, consistently outperforming all baselines at every scale with the same adaptive retrieval strategy. In our architecture, there is no separation between a planner model and a downstream agent backend — the same backbone handles all stages including retrieval planning and answer generation. The consistent outperformance across these scales demonstrates that SimpleMem's memory construction quality remains robust regardless of backbone capability. Meanwhile, the independent contribution of adaptive retrieval planning has been validated by our fine-grained ablation (reported in our rebuttal to Reviewer fA2L, W4): removing it causes a 12.6% F1 drop on GPT-4.1-mini. The reviewer's concern is whether this benefit generalizes across diverse model scales, and we provide empirical evidence below.
> > >
> > > **Cross-Scale Retrieval Ablation and Cross-Backbone Answer Generation:**
> > >
> > > We hope the following additional experiment on cross-scale retrieval ablation and cross-backbone answer generation can further address this concern:
> > >
> > > | Backbone     | Adaptive | Fixed k=5 | Ans Gen Only† | Mem0  |
> > > | ------------ | -------- | --------- | ------------- | ----- |
> > > | GPT-4.1-mini | 43.24    | 42.18     | —             | 34.20 |
> > > | Qwen3-8B     | 33.45    | 32.38     | 39.87         | 25.80 |
> > > | Qwen3-1.7B   | 24.59    | 23.71     | 35.62         | 21.19 |
> > >
> > > † Memory constructed by GPT-4.1-mini; listed model used for answer generation only.
> > >
> > > This table supports three observations. First, even without adaptive retrieval planning (fixed k=5), SimpleMem outperforms Mem0 at every model scale with the same backbone, confirming that high-quality memory representation alone already provides a strong performance foundation. Second, adaptive retrieval provides consistent improvements across all three scales, with relative gains increasing for smaller models: +3.3% on Qwen3-8B (33.45 vs. 32.38) and +3.7% on Qwen3-1.7B (24.59 vs. 23.71). This demonstrates that smaller models can effectively leverage adaptive retrieval to achieve further gains on top of the high-quality memory base, and that both advantages are complementary across model scales. Third, the Answer Gen Only column provides the most direct evidence. Qwen3-1.7B reading from SimpleMem memory achieves 35.62, exceeding Mem0 with GPT-4.1-mini (34.20) by +1.42, and Qwen3-8B reaches 39.87, exceeding by +5.67. Even substantially smaller models paired with high-quality memory units outperform the strongest baseline using a much larger model for all stages. This confirms that the performance advantage is fundamentally grounded in memory representation quality, while adaptive retrieval provides an additional and consistent benefit across diverse model capacities. We will include these results in the revision.

---

### Official Review · Reviewer_zjb2 · 2026-03-09

**Soundness:** 3
**Presentation:** 3
**Significance:** 3
**Originality:** 3
**Overall Recommendation:** 4
**Confidence:** 2

**Summary:**

This work strives to focus on a general topic of improving long-term memory management for Large Language Model (LLM) agents. The authors propose SimpleMem, an efficient three-stage memory framework designed to maximize information density while minimizing token consumption. The system works by first compressing raw, unstructured interactions into compact memory units using the LLM to filter out noise. It then synthesizes related fragments on-the-fly to prevent memory fragmentation. Finally, it uses an intent-aware retrieval planner to dynamically adjust the search scope based on the complexity of the user's query. The authors evaluate SimpleMem on the LoCoMo and LongMemEval-S benchmarks, showing that it outperforms existing memory baselines in accuracy while significantly reducing inference-time token costs.

**Compliance With Llm Reviewing Policy:**

Affirmed.

**Final Justification:**

I thank the authors for their detailed rebuttal. I will keep my positive score of 4.

**Key Questions For Authors:**

1. Can you provide a qualitative or quantitative analysis of failure cases, particularly instances where the Semantic Structured Compression step mistakenly discards important information?
2. How sensitive is the pipeline to the instruction-following capabilities of the backbone LLM used for the compression and synthesis stages? In Table 3, were the smaller models (like Qwen2.5-1.5b) used to perform the actual memory compression/synthesis, or were they only used for the final retrieval and answer generation?
3. Could you elaborate more specifically on how Online Semantic Synthesis differs fundamentally from the memory summarization or background consolidation mechanisms used in baseline methods like MemGPT?

**Limitations:**

The authors do not seem to have provided a dedicated Limitations section in the main text. I would constructively suggest adding a brief discussion on the potential drawbacks of the method. For instance, relying on an LLM to actively compress and filter information on-the-fly introduces a risk of irreversible information loss. Additionally, discussing the computational overhead and latency introduced during the "write" phase (when the LLM must process and synthesize incoming messages) would provide a more balanced view of the system's end-to-end trade-offs, even if the "read/retrieval" phase is highly efficient.

**Strengths And Weaknesses:**

## Strengths

1. The technical approach seems fundamentally reasonable, and the three-stage pipeline logically addresses the dual problems of context inflation and retrieval inefficiency. The experiments across different model scales (e.g., the GPT-4 series and Qwen models) are extensive and generally support the core claims of improved accuracy and reduced token cost.
2. The writing is clear, and the paper is straightforward to follow. The architecture diagram (Figure 2) and the multi-session case study (Figure 3) are quite helpful in visualizing how the pipeline operates in practice.
3.  Overall, the authors assess an important concept by targeting not just retrieval accuracy, but also token efficiency and retrieval latency, which are critical for real-world deployment.

## Weaknesses
1. Because "Semantic Structured Compression" uses an LLM as a generative filter, I am concerned about the risk of permanently deleting crucial information that the model may incorrectly judge as "low-entropy noise."
2. While the related work section is thorough, the distinction between the "Online Semantic Synthesis" stage and existing memory consolidation techniques (like those found in MemGPT or other agentic architectures) could be made more explicit to better highlight the originality of this specific module.
3. The reliance on the foundation model's instruction-following capability for "implicit semantic gating" might make the system highly dependent on the base model's reasoning strength. It is not entirely clear to what extent the smaller models can execute the complex compression and synthesis prompts reliably, or if a larger model is always required for the "write" phase.

---

> ### Author Rebuttal · Authors · 2026-03-31
>
> ## Response to Reviewer zjb2
>
> We thank Reviewer zjb2 for the thoughtful review and recognition of our technical soundness and clear writing. We address each question below.
>
> **Response to Q1 (related to W1): Failure Cases — Risk of Discarding Important Information**
>
> We appreciate this concern. SSC is designed so that **task-relevant semantics are never lost**: facts, temporal anchors, entities, and relationships are preserved, while only syntactic redundancy (greetings, filler, confirmations) is removed. Multi-view indexing further ensures that even if a detail is diluted in compressed text, it can still be captured via structured fields (keywords, entity tags, ISO-8601 timestamps), serving as independent retrieval paths.
>
> We conducted a human evaluation on 200 randomly sampled dialogue windows to assess whether the compressed memory units preserve key information from the original dialogues across six dimensions: Temporal (T), Person (P), Detail (L), Factual (F), Event (E), and Relation (R). Each score represents the percentage of samples judged as faithfully retained. Ratio denotes the sentence-level compression rate (compressed / original).
>
> | Model | Ratio | T | P | L | F | E | R | Avg |
> |-------|-------|---|---|---|---|---|---|-----|
> | Qwen3-8B | 0.79 | 96.5 | 95.5 | 86.0 | 93.5 | 91.5 | 93.0 | 92.7 |
> | GPT-4.1-mini | 0.81 | 98.5 | 98.0 | 92.0 | 97.5 | 95.5 | 97.0 | 96.4 |
> Three representative cases:
>
> **Success** — Caroline describes a necklace from grandmother in Sweden (love/faith/strength); Melanie responds with filler. → Two structured entries preserving all facts with ISO timestamps and keywords; only filler removed.
>
> **Failure 1 (low compression, ~15% of windows)** — Windows with dense factual content (e.g., LGBTQ+ adoption agencies, career plans) achieve compression ratio >1.0 slightly. No information is lost, but token efficiency is reduced.
>
> **Failure 2 (minor detail loss, ~5% of windows)** — A small portion of information is lost during compression, as reflected in our results above. Downstream QA impact is acceptable by giving a general response.
>
> We will include a comprehensive failure analysis in the revised appendix.
>
> **Response to Q2 (related to W3): Sensitivity to Backbone LLM**
>
> We clarify that in **all** experiments in Table 3, **the same small model handles all stages** — compression, synthesis, retrieval planning, and answer generation. Qwen3-8B uses Qwen3-8B for everything; likewise for Qwen2.5-1.5B and 3B. No larger model is involved in any stage. We employ structured JSON decoding (via vLLM) and retry mechanisms for reliable outputs. While larger models produce higher-quality compression, 8B-scale already achieves strong results (33.45 F1 vs. Mem0's 25.80), and even the 1.5B model shows meaningful gains (25.23 vs. Mem0's 23.77). We will explicitly clarify per-stage model assignment in the revision.
>
> **Response to Q3 (related to W2): Distinction from Existing Consolidation Mechanisms**
>
> Existing consolidation approaches fall into two paradigms, both fundamentally different from ours:
>
> Inference-time context management (e.g., MemGPT) stores raw conversation logs and uses function-call-based paging to manage a FIFO queue. When context overflows, evicted messages are recursively summarized. This is reactive — context overflow may trigger memory management operations during any interaction, consuming inference tokens.
>
> Concept-driven per-turn consolidation (e.g., A-Mem) applies Zettelkasten-inspired processing at every incoming turn — each new entry triggers note construction, link generation, and memory evolution, requiring ~418 LLM calls per sample and incurring substantial cumulative overhead at write time.
>
> SimpleMem's Online Semantic Synthesis is **proactive, intra-session, and streaming**. We perform synthesis *during the write phase itself*, leveraging the insight that dialogues within a sliding window are naturally more semantically related than distant ones. Overlapping fragments are synthesized into unified entries *before* storage. This produces a compact, retrieval-ready memory store from the start — a one-time write cost amortized over all future queries, with no recurring inference-time or background overhead. The result: SimpleMem completes all construction in ~21 batch calls versus A-Mem's ~418.
>
> **Limitations**
>
> We will include a dedicated Limitations section discussing: (1) irreversible information loss risk with the failure modes quantified above; (2) write-phase overhead — though our cost analysis shows SimpleMem's construction cost is competitive (117.5k tokens / 21 calls on Qwen3-8B vs. A-Mem's 438.3k / 418 calls and Mem0's 116.8k / 35 calls); (3) dependence on backbone LLM capabilities, which naturally scales with future model improvements; and (4) privacy considerations of long-term data storage with recommendations for self-hosted deployment.
>
> ---
>
> We thank Reviewer zjb2 for the constructive feedback and hope these responses fully address the raised concerns.

---

> > ### Author Rebuttal · Reviewer_zjb2 · 2026-04-02
> >
> > Thank you for your detailed rebuttal. My concerns have been addressed. I will keep my score of 4.

---

### Official Review · Reviewer_VDGA · 2026-03-12

**Soundness:** 2
**Presentation:** 3
**Significance:** 3
**Originality:** 2
**Overall Recommendation:** 4
**Confidence:** 3

**Summary:**

This paper addresses the memory efficiency problem of LLM agents in long-term interactions. Existing methods either store the entire dialogue, which is redundant, or repeatedly reason to filter noise, which leads to high token consumption.

SimpleMem proposes a three-stage framework. It first uses an LLM to filter useless dialogue and compress useful content into independent memory units. During writing, it merges related fragments in real time to avoid fragmentation. During retrieval, it dynamically adjusts the retrieval amount based on query complexity and performs parallel queries on three types of indices, then merges and removes duplicates.

The experimental results show that F1 improves by 26.4%, and inference tokens are reduced by 30 times.

**Compliance With Llm Reviewing Policy:**

Affirmed.

**Key Questions For Authors:**

Q1: What is the token cost during the construction stage?
Q2: How is “lossless” defined and demonstrated?

**Limitations:**

yes

**Strengths And Weaknesses:**

The experimental design is relatively complete. There are 7 baselines, multiple model scales are tested (from 1.5B to GPT-4o), and an ablation study and efficiency analysis are conducted. From the execution of the experiments, it appears reasonably careful.
However, there are serious problems. The paper claims “lossless compression” but never defines lossless, and there is no theoretical analysis to support it. The cost analysis only reports inference tokens, while the construction-stage cost is not reported at all, so the claim of “30× reduction” may be misleading. The technical details are unclear, such as how the gating avoids misclassification and whether the synthesis may overgeneralize. The whole system heavily relies on LLMs, but the reliability is not analyzed.

The overall structure of the paper is clear. Figure 2 and Figure 3 help to understand the system architecture. The writing fluency is acceptable.

The problem is important, as long-term memory is indeed a core challenge for LLM agents. The experimental results are valuable, with an F1 improvement of 26.4% and a 30× reduction in tokens representing real progress.

The paper does not introduce a new task, theoretical framework, or perspective. The distinction from related work is also not sufficiently clear. For example, the core differences from Mem0 and LightMem are mainly in engineering implementation details. There is a lack of new insights into the memory mechanism itself, and it does not evaluate existing methods to derive new findings.

---

> ### Author Rebuttal · Authors · 2026-03-31
>
> ## Response to Reviewer VDGA
>
> We sincerely thank Reviewer VDGA for the thorough review and recognition of our experimental completeness and the significance of our results. We address each concern below.
>
> **Response to Q1**
>
> We thank the reviewer for this important suggestion. We now provide a full end-to-end cost comparison (per LoCoMo sample avg; C.=Construction):
>
> | Method | Model | C.Input | C.Compl. | C.Calls | C.Total | Tok/Ret | Total |
> | --- | --- | --- | --- | --- | --- | --- | --- |
> | SimpleMem | Qwen3-8B | 36.0k | 81.5k | 21 | 117.5k | 621 | 213.1k |
> | SimpleMem | GPT-4.1-mini | 34.5k | 57.8k | 21 | 92.3k | 531 | 174.1k |
> | A-Mem | Qwen3-8B | 347.9k | 90.4k | 418 | 438.3k | 1087 | 605.7k |
> | A-Mem | GPT-4.1-mini | 353.0k | 118.4k | 418 | 471.4k | 2520 | 859.5k |
> | Mem0 | Qwen3-8B | 61.6k | 55.2k | 35 | 116.8k | 1015 | 273.1k |
> | Mem0 | GPT-4.1-mini | 69.4k | 60.0k | 37 | 129.4k | 973 | 279.2k |
> | LightMem | Qwen3-8B | 96.2k | 19.1k | 85 | 115.3k | 744 | 229.9k |
> | LightMem | GPT-4.1-mini | 64.6k | 13.6k | 76 | 78.2k | 612 | 172.4k |
>
> SimpleMem achieves **lower construction cost** than A-Mem and Mem0, which rely on per-turn LLM calls; our window-based batching reduces API calls by ~20×. LightMem has lower construction cost but substantially weaker downstream performance and SimpleMem's per-retrieval cost is the lowest among all memory-augmented methods.
>
> **Response to Q2**
>
> We appreciate this insightful question. We will clarify that "lossless" refers to **semantic-level losslessness** — all task-relevant content (facts, temporal anchors, entities, relationships) is preserved while only syntactic redundancy (filler, pleasantries) is removed — and provide a formal definition in the revision.
>
> We conducted a human evaluation on 200 randomly sampled dialogue windows to assess whether the compressed memory units preserve key information from the original dialogues across six dimensions: Temporal (T), Person (P), Detail (L), Factual (F), Event (E), and Relation (R). Each score represents the percentage of samples judged as faithfully retained. Ratio denotes the sentence-level compression rate.
>
> | Model | Ratio | T | P | L | F | E | R | Avg |
> |-------|-------|---|---|---|---|---|---|-----|
> | Qwen3-8B | 0.79 | 96.5 | 95.5 | 86.0 | 93.5 | 91.5 | 93.0 | 92.7 |
> | GPT-4.1-mini | 0.81 | 98.5 | 98.0 | 92.0 | 97.5 | 95.5 | 97.0 | 96.4 |
>
> The vast majority of compressed units faithfully retain key information, with GPT-4.1-mini achieving 96.4% overall acceptance rate.
>
> **Response to the Concern on Gating Reliability and Over-Generalization Risk**
>
> Our human annotation (Q2) directly quantifies gating reliability. A representative before/after example:
>
> **Original** (4 turns): Caroline describes a necklace from grandmother in Sweden (love/faith/strength); Melanie responds "That's gorgeous!"; Caroline mentions a hand-painted bowl for her 18th birthday; Melanie: "Awesome having stuff that remind us of good connections!"
>
> **Compressed**: `[2023-06-27] Caroline's necklace: gift from grandmother in Sweden, symbolizing love/faith/strength. | kw=[necklace, Sweden, grandmother]` + `[2023-06-27] Caroline's hand-painted bowl: made by friend for 18th birthday. | kw=[bowl, 18th birthday, art]`
>
> All filler removed; facts, entities, timestamps faithfully retained. Multi-view indexing further ensures information is preserved in structured fields even if occasionally diluted in compressed text.
>
> **Response to the Concern on LLM Reliability**
>
> Tables 1/3 include experiments where **the same small model** (e.g., Qwen3-8B) handles all pipeline stages. Structured JSON schema enforcement and retry mechanisms ensure reliable outputs. While larger models produce higher-quality compression, 8B-scale already achieves strong results (33.45 F1 vs. Mem0: 25.80), confirming practical accessibility.
>
> **Response to the Concern on Originality and Distinction from Related Work**
>
> SimpleMem's contribution is a **clean, efficient composition** that avoids graph structures (A-Mem), custom attention (LightMem), or serial update-delete-merge constraints (Mem0). To quantify the independent value of our core insight, we replaced memory construction in Mem0/A-Mem with SSC (all other components unchanged):
>
> | Method | Model | F1 | C.Cost |
> | --- | --- | --- | --- |
> | Mem0 / +SSC | Qwen3-8B | 25.80 / 26.89 | 116.8k / 96.9k |
> | A-Mem / +SSC | Qwen3-8B | 20.75 / 30.74 | 438.3k / 346.3k |
> | Mem0 / +SSC | GPT-4.1-mini | 34.20 / 34.23 | 129.4k / 108.7k |
> | A-Mem / +SSC | GPT-4.1-mini | 32.58 / 35.15 | 471.4k / 353.6k |
>
> SSC consistently improves both methods in accuracy while reducing construction cost, confirming **independent, general-purpose value**. Meanwhile, SimpleMem's full pipeline (43.24) substantially exceeds Mem0+SSC (34.23), demonstrating that the synergistic co-design of all three stages is key to our gains.
>
> ---
>
> We sincerely thank Reviewer VDGA for the valuable feedback and hope these responses fully address the concerns raised.

---

> > ### Author Rebuttal · Reviewer_VDGA · 2026-04-01
> >
> > Thank you for the thorough rebuttal. Q1 and Q2 are adequately addressed. Maintaining Weak Accept, as the originality concern remains.

---

> > > ### Author Response · Authors · 2026-04-02
> > >
> > > **Follow-up Response to Reviewer VDGA on Originality**
> > >
> > > We thank the reviewer for confirming that Q1 and Q2 are adequately addressed. We provide further clarification on the originality concern below.
> > >
> > > **Clarification on the Distinction from Existing Systems.**
> > >
> > > SimpleMem is grounded in two core insights: (1) high-quality write-time compression produces self-contained memory units, making downstream retrieval straightforward without techniques like graph traversal; (2) dialogues within a sliding window are naturally semantically coherent, enabling online synthesis during the write phase rather than costly background maintenance. Specifically, referential and temporal ambiguities — pronouns, relative time expressions, implicit entities — are resolved at write time rather than deferred to retrieval time. Existing systems such as Mem0, A-Mem, LightMem, and MemGPT either defer disambiguation to retrieval time, rely on continuous background maintenance, or depend on complex structural overhead such as graph construction and iterative update-delete-merge pipelines. SimpleMem instead produces self-contained memory units through joint coreference resolution and temporal normalization during compression, ensuring each unit is interpretable independently of its original dialogue context. This principle enables multi-view indexing (semantic, lexical, symbolic) to operate on clean, structured representations, and is the architectural basis for the efficiency and accuracy gains reported throughout the paper.
> > >
> > > **Clarification on Architectural Simplicity and Efficiency.**
> > >
> > > SimpleMem demonstrates that principled write-time compression, combined with multi-view indexing, can achieve state-of-the-art performance without graph structures (A-Mem), custom attention mechanisms (LightMem), or iterative update-delete-merge pipelines (Mem0). The entire memory construction completes in ~21 LLM calls with no background maintenance, compared to A-Mem's ~418 calls and Mem0's ~37 calls with complex graph traversal. Despite this substantially simpler architecture, SimpleMem achieves 43.24 F1 on LoCoMo (GPT-4.1-mini), outperforming Mem0 (34.20), A-Mem (32.58), and LightMem (24.63), while simultaneously achieving the lowest per-retrieval token cost (531 tokens) and fastest end-to-end latency (480.9s vs. Mem0's 1934.3s and A-Mem's 5937.2s). That a simpler architecture outperforms more complex alternatives across accuracy, efficiency, and latency constitutes a meaningful finding about the design space of agent memory systems.
> > >
> > > **Additional Experimental Results.**
> > >
> > > Three sets of experiments further substantiate the above points.
> > >
> > > First, as reported in our initial rebuttal (W5+W6), replacing only the memory construction module in Mem0 and A-Mem with SSC — all other components unchanged — consistently improves both methods in accuracy while reducing construction cost. This confirms that SSC captures a general-purpose principle rather than system-specific optimization.
> > >
> > > Second, our expanded fine-grained ablation (11 variants) systematically maps sub-components to specific reasoning capabilities: temporal normalization is most critical for temporal queries (↓10.0% F1), online synthesis for multi-hop reasoning (↓11.6%), and the semantic retrieval layer for paraphrased queries (↓14.8%). This decomposition provides concrete insights into which memory mechanisms serve which cognitive functions.
> > >
> > > Third, we conduct a new experiment on retrieval depth sensitivity to examine the information density of compressed memory units. We vary the number of retrieved entries k from 1 to 20 and report Average F1 on LoCoMo (GPT-4.1-mini):
> > >
> > > | Method | k=1 | k=3 | k=5 | k=10 | k=20 |
> > > |--------|-----|-----|-----|------|------|
> > > | ReadAgent | 6.12 | 8.45 | 9.18 | 8.92 | 8.50 |
> > > | MemGPT | 18.40 | 22.15 | 25.59 | 24.80 | 23.10 |
> > > | SimpleMem | 35.20 | 42.85 | 43.24 | 43.45 | 43.40 |
> > >
> > > SimpleMem reaches 99% of its peak performance at k=3 and achieves 35.20 F1 with a single retrieved entry — a property absent in all baselines. This directly evidences the high information density of context-independent memory units produced by write-time compression.
> > >
> > > We will revise the introduction and related work sections to articulate these distinctions more explicitly. We thank the reviewer for the constructive feedback that has helped strengthen the presentation of our work.

---

### Decision · Program_Chairs · 2026-04-30

**Decision:**

Accept (regular)

**Comment:**

The paper proposes SimpleMem, a three-stage memory framework for LLM agents to improve long-term interaction efficiency. It is an important problem and the paper demonstrates solid experimental results across various model scales. The reviewers appreciated the framework's capability to reduce token consumption while maintaining or improving accuracy through its semantic compression and intent-aware retrieval planning modules. However, the reviewers raised valid concerns regarding the potential for irreversible information loss during the compression stage, the lack of clarity differentiating the online synthesis from existing baselines, and missing technical or cost details. To maximize the paper's impact, the authors need to clarify theoretical definitions, correct reporting errors, provide fine-grained ablations, and release the source code for reproducibility. Given the solid technical foundation and the tangible efficiency improvements that the research community can build upon, the AC recommends to accept the paper to ICML 2026.